# Telomere DNA length regulation is influenced by seasonal temperature differences in short-lived but not in long-lived reef-building corals

Telomeres are environment-sensitive regulators of health and aging. Here, we present telomere DNA length analysis of two reef-building coral genera revealing that the long- and short-term water thermal regime is a key driver of between-colony variation across the Pacific Ocean. Notably, there are differences between the two studied genera. The telomere DNA lengths of the short-lived, more stress-sensitive *Pocillopora* spp. colonies were largely determined by seasonal temperature variation, whereas those of the long-lived, more stress-resistant *Porites* spp. colonies were insensitive to seasonal patterns, but rather influenced by past thermal anomalies. These results reveal marked differences in telomere DNA length regulation between two evolutionary distant coral genera exhibiting specific life-history traits. We propose that environmentally regulated mechanisms of telomere maintenance are linked to organismal performances, a matter of paramount importance considering the effects of climate change on health.

At the natural ends of chromosomal DNA, telomeres protect against unwanted activation of the DNA damage response, help organize chromosomes, and regulate gene expression[1]. These functions rely on the specific chromatin structures[2] assembled around arrays of simple DNA repeats, which can vary in sequence among species (e.g., TTAGGG in vertebrates, as well as cnidarians)[3]. Importantly, the telomere-specific chromatin structure depends on the number of these repeats, such that telomere DNA length (TL) is a crucial determinant of genome maintenance and function.

TL is regulated by the balance between the pathways associated with telomere DNA elongation and erosion[1,4]. Elongation is usually mediated by a special reverse transcriptase called telomerase, or, less commonly, through recombination. Meanwhile, erosion is caused by the replication of DNA ends, nucleases, and recombinases. In the somatic cells of several vertebrates, telomerase expression is down-regulated at the end of embryogenesis, leading to progressive TL shortening throughout the life course. In other organisms, telomerase expression is maintained throughout life[5-7], and TL does not decrease

with age[8]. In many invertebrates and aquatic vertebrates, the persistent activity of telomerase in somatic tissues may be associated with their high regenerative potential.

TL regulation is complex. For example in budding yeast, TL is directly or indirectly modulated by more than 400 genes[9,10] and is also influenced by non-genetic factors such as oxidative stress and a variety of environmental factors[11]. A wealth of studies shows that different types of stressful conditions, e.g., elevated temperature, lead to TL shortening[12,13]. Overall, TL maintenance regulation is determined by a combination of genetic, biotic, and abiotic factors that influence the pathways responsible for telomere elongation and shortening.

TL differs greatly between taxa, individuals, and ecoregions[14-23]. The significance of these variations is under intense investigation, particularly how they can determine life-history traits and health and whether they have adaptive values to cope with physiological and environmental constraints[24-29]. In the general human population, TL is correlated with lifespan and the risk of developing various age-related diseases[30-33]. Moreover, excessive TL shortening is known to be the

✉ e-mail: Alice.Rouan@outlook.fr; Eric.Gilson@univ-cotedazur.fr

mechanism behind rare genetic diseases known as "premature aging syndrome"[34]. Similarly, to humans, in different organisms, associations of short TL with decreased lifespan and higher mortality risk were reported[15,35–38]. A causal relationship between short TL, health and longevity is shown in genetic models of plant, yeast, nematode, killifish, zebrafish, and mouse[39–44]. Despite all these findings supporting the view that short TL has deleterious effects, the interspecific variations of lifespan cannot be simply explained by differences in TL[24]. Even the concept of the deleterious effects of short TL must be revisited since there are examples of higher survival correlating with short telomeres[45,46], suggesting the possibility of advantageous effects of short TLs, such as preventing cancer formation or being prone to activate metabolic and survival pathways[20,47,48].

The somewhat disparate range of findings reported above make it necessary to investigate beyond simple TL to understand the mechanisms linking telomeres and organismal performance. This is how it was found that it is rather the rate of TL shortening than absolute mean TL that correlates with lifespan in different organisms[49–51]. Interestingly, in both yeast and mice, there are indications that higher or aberrant rates of telomere erosion increase the

percentage of extremely short telomeres[52,53], which in turn can trigger cellular senescence even when they are in limited numbers[54,55]. Telomeric structures other than TL are certainly also to be taken into consideration, like the presence of a gain-of-function variant of the shelterin *TERF1* gene in the long-lived naked mole rat[28]. Overall, the impact of telomere regulation on physiology and life-history traits remains a very open question.

In this study, we addressed the question of whether mechanisms regulating TL evolve differently in organisms acquiring specific life-history traits. For this, we used reef-building corals as model organisms. Indeed, these animals are particularly suited here due to their ectothermic metabolism, which makes them more plastic to environmental factors, including seawater temperatures, their sessile lifestyle in the adult stage, which prevents escape from environmental stress, allowing to track the effects of past events, and their broad range of life-history traits, even within the same habitat[56]. For example, the TL of the reef-building coral *Stylophora pistillata* changes in response to dark-induced bleaching, supporting a role for coral telomeres in stress responses[57]. Moreover, understanding the role of telomeres in the

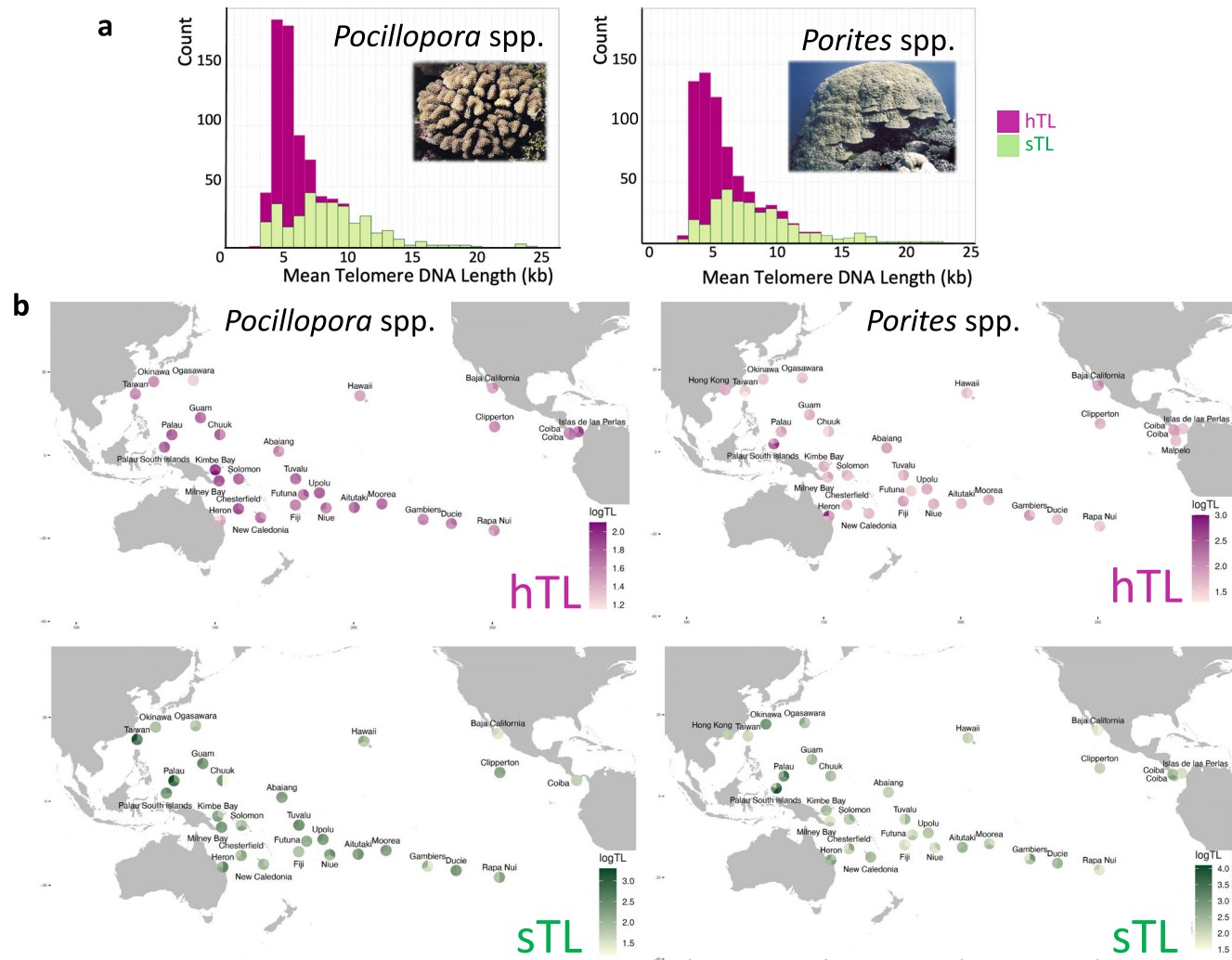

**Fig. 1 | Telomere DNA length (TL) variation among two genera of reef-building coral in 32 islands of the Pacific Ocean. a** Mean Telomere DNA length TL (in kilobases) distribution of the host (hTL, purple) and their symbionts (sTL, green) for the *Pocillopora* spp. samples (left) and the *Porites* spp. *samples (right)*. **b** Maps of log-transformed mean TL averaged at the sampling site level for the host (hTL, purple) and their symbionts (sTL, green), for the *Pocillopora* spp. samples (left) and the *Porites* spp. *samples (right)*. The results are displayed as pie-charts with each sampling site for a given island is represented as a slice. Island names are displayed, and pie-charts have been displayed to avoid overlap. Source data are provided as a Source Data file.

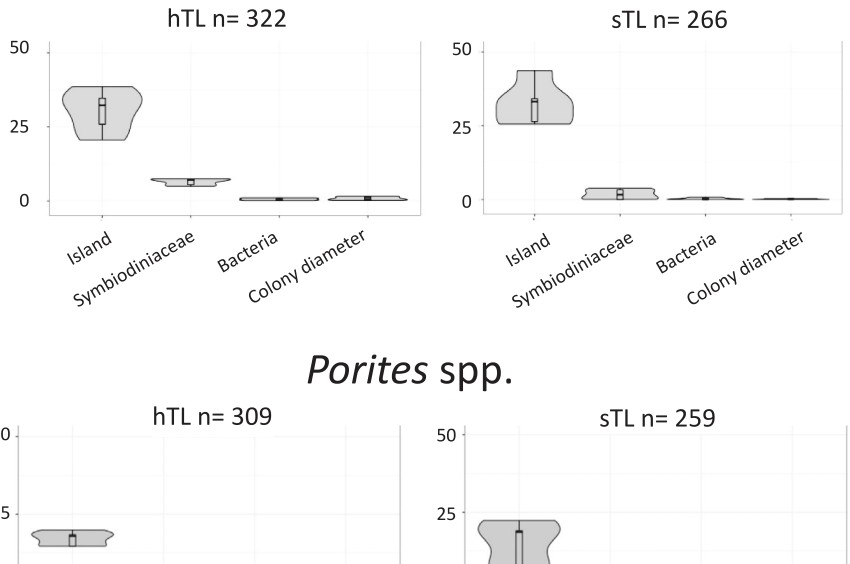

**Fig. 2 | Telomere DNA length (TL) variation is determined by island.** Percentage of TL variation explained by the island of origin, Symbiodiniaceae and bacterial communities, and colony morphology (diameter) for *Pocillopora* spp. and *Porites* spp. colonies across 32 islands of the Pacific Ocean. In this analysis, i.e., mean, median, Q1, Q3 and IQ were used. n refers to the number of samples. The boxplots are defined as follows: the lower and upper bounds of the box represent the first (Q1) and the third (Q3) quartile, respectively. The entire box represents the inter-quartile range (IQ). The median is represented as a line across the box. Whiskers extending from Q1 and Q3 are defined as 1.5xIQ. Source data are provided as a Source Data file.

health of reef-building corals in a warming ocean is an important, yet unexplored question.

We present here the results of a comparative study between two evolutionary distant reef-building coral genera residing in a gradient of environmental conditions across the Pacific Ocean. This large geographical range, high resolution study used the coral samples and the vast multi-level dataset generated during the *Tara* Pacific expedition[58,59], which targeted the massive *Porites* spp. that are slow-growing, stress-resistant, and long-lived, with estimated life expectancies of >600 years, and the branched *Pocillopora* spp. that are fast-growing, sensitive to bleaching, and short-lived with estimated average colony ages of a few decades[56,60]. We found that TL variation was largely explained by historical patterns of sea surface temperatures (SST) in both coral genera, albeit with marked differences. *Pocillopora* spp. telomeres were sensitive to seasonal temperature variation, whereas *Porites* spp. TL was slightly positively correlated with the presence/prevalence of past heat waves. These results reveal complex relationships between telomere and environment that go beyond the current view that TL decreases upon environmental stress. Considering the different life-history traits of these two coral genera and the known functions of telomeres, we propose that specific mechanisms regulating TL in response to environmental variations contribute to the different longevity and stress-resistance properties encountered among reef-building corals.

## Results

The measures and analyses of coral TL benefited from the *Tara* Pacific expedition (2016–2018) that studied coral reefs around 32 islands across the Pacific Ocean[58]. The complete description of the coral sampling methodology and the access to the different datasets generated from this expedition are presented in ref. 59 while the description of the workflow for multi-omics data is given in ref. 61. The

reef-building corals studied here were collected according to their resemblances to the branched *Pocillopora meandrina* (named *Pocillopora* spp. hereafter) and to the massive *Porites lobata* (named *Porites* spp. hereafter). *Porites* spp. and *Pocillopora* spp. correspond to two evolutionary distant coral genera widely distributed across the Pacific Ocean[62].

### Coral telomere DNA length variation across the Pacific Ocean

In order to measure TLs in the coral samples, we used the gold standard procedure named TRF (for Terminal Restriction Fragment), previously set up in *Stylophora pistillata* to measure sequentially the DNA lengths of the coral host (TTAGGG)n (hereafter hTL) and symbiont (TTTAGGG)n (hereafter sTL)[57]. To avoid any noise from the first probing every membrane was controlled after the stripping step with an overnight exposure to phosposcreen[57]. The terminal localization of the radiolabeled fragments was checked by Bal31 digestion (Supplementary Fig. 1a). We were also concerned by variations according to the localization of the sample within the colony[60,63]. Thus, we analyzed a dedicated sampling at the Clipperton island by cutting the branches of six colonies of *Pocillopora* spp. into apex and base segments and four massive colonies of *Porites* spp. sampled at their top (apex) and bottom (base) parts. We did not find a statistically significant difference in TL between the apex and the base for both genera samples (Supplementary Fig. 1b). We observed a slight trend toward shorter telomeres at the apex of *Pocillopora* spp. branches. Nevertheless, we took the precaution to measure TL at the apex of the *Pocillopora* spp. samples.

The results of 851 colonies collected from 99 reef sites around 32 islands spanning nearly 17 000 km overwater distance across the Pacific Ocean (443 colonies of *Pocillopora* spp. and 408 colonies of *Porites* spp.) are shown in Fig. 1a, b and Supplementary Figs. 2–4 (the full TL dataset is given in ref. 64). Beyond mean TL, we calculated

several parameters describing the shape of the TL distribution (an eventual skewness toward short or long TLs): median (med), 1st quartile TL limit (Q1), 3rd quartile TL limit (Q3) and interquartile distance (IQ) (Supplementary Fig. 2). The mean TLs were further analyzed in this study, unless otherwise stated.

Our results revealed mean TL variations among colonies (Fig. 1a, b, Supplementary Figs. 3, 4). The hTL distribution is more centered than the widely spread variation of sTL (Fig. 1a). The averages of all colonies were about twice as long for sTL (9.2 kb for *Pocillopora* spp.; 11.5 kb for *Porites* spp.) as for hTLs (5.1 kb for *Pocillopora* spp.; 5.8 kb for *Porites*). The range of hTLs for both genera is consistent with previous reports in cnidarian[57,65–67].

### Weak (*Pocillopora* spp.) and no (*Porites* spp.) correlation between telomere DNA length variation and colony size

Although coral colony size is considered a poor predictor of genetic age[68], especially for species prone to frequent breakage and regrowth[69], it may be used as a proxy for cumulative growth or colony age[70]. Thus, if the TL of the coral colony shortened with cumulative growth or age as in several non-colonial metazoan organisms[8,51], one would expect a relationship between TL and the size of the colony. Using regression models, we evaluated associations between TLs and colony diameter estimated from pictures taken at the time of sampling[71]. Linear, quadratic, and logarithmic regression models were tested. Significant models were found for *Pocillopora* spp., explaining at best 9% of the hTL and sTL variation by combining log(TL) with a linear effect of Colony Diameter Mean for both hTL and sTL (Supplementary Table 1) corresponding to a slightly negative slope (Supplementary Fig. 5a). For *Porites* spp., none of the models were significant ($p$-value > 0.05 with a slope of −0.0057 for hTL and 0.014 for sTL) (Supplementary Fig. 5b).

To rule out any confounding effect of the site or island of origin, we sampled additional colonies at the same reef sites according to four diameter classes: one site at Clipperton Island for 13 *Pocillopora* spp. colonies and 14 *Porites* spp. colonies, and one site for 12 *Porites* spp. colonies in Palau. No difference in mean hTL among the different classes of colony size was detected (Supplementary Fig. 5c-f).

Overall, these results indicate a slight difference in the association of TL to cumulative growth or age between the two genera: a 9% colony size effect can be modeled for *Pocillopora* spp. colonies while no significant effect can be detected for *Porites* spp. colonies. The robustness of this conclusion stems from the large geographical distance and size classes sampled. For instance, the age range of sampled *Porites* colonies can be estimated between 20 and 600 years assuming an average growth rate of 0.9 cm/year[72]. Moreover, since the *Porites* spp. colonies are not subjected to frequent fragmentations[69], as confirmed by an absence of genetic clones among a subset of sampled *Porites* spp. colonies[71], one expects a good relationship between their size and genetic age.

### Driving effect of island of origin on telomere DNA length variation

Next, we performed variance partitioning analysis, which uses a linear mixed model to partition the variance attributable to multiple variables in the data, and linear models to assess the relative contributions to TL variation of the island of origin, the Symbiodiniaceae composition (47 different profiles[73], see the Methods section), the bacterial community composition (7 different profiles for *Pocillopora* spp. and 8 for *Porites* spp[74].) and the colony diameter[71].

Using variance partitioning, for the two genera, the main driver of mean hTL and sTL variance was the island of origin (Fig. 2, the marginal and conditional $R^2$ values are given in Supplementary Table 2). The effect of colony diameter on *Pocillopora* spp. hTL variation was barely detectable (0.69% of the explained variance), which is even much lower than the weak one detected by the regression model (see above).

**Table 1 | Commonality analysis for the best regression models describing TL for the 32 islands**

| Variable | Coefficient | Total % |
|---|---|---|
| *Pocillopora* hTL: log(TL_Mean) ~ island + Symbiodinicaeae + bacteria ($R^2$ adj = 0.43) | | |
| Unique to island | 0.2453 | 48.11 |
| Unique to Symbiodiniaceae | 0.0684 | 13.41 |
| Unique to bacteria | 0.0242 | 4.74 |
| Common to island, and Symbiodiniaceae | 0.1324 | 25.96 |
| Common to island, and bacteria | 0.0312 | 6.11 |
| Common to Symbiodinicaeae, and bacteria | −0.0106 | −2.07 |
| Common to island, Symbiodiniaceae, and bacteria | 0.0191 | 3.74 |
| Total | 0.5099 | 100 |
| *Pocillopora* sTL: log(TL_Mean) ~ island + Symbiodiniaceae ($R^2$ adj = 0.55) | | |
| Unique to island | 0.2897 | 47.33 |
| Unique to Symbiodiniaceae | 0.0387 | 6.32 |
| Common to island, and Symbiodiniaceae | 0.2837 | 46.34 |
| Total | 0.6121 | 100 |
| *Porites* hTL: log(TL_Mean) ~ island ($R^2$ adj = 0.29) | | |
| *Porites* sTL: log(TL_Mean) ~ island + Symbiodiniaceae ($R^2$ adj = 0.33) | | |
| Unique to island | 0.2732 | 62.46 |
| Unique to Symbiodiniaceae | 0.065 | 14.85 |
| Unique to bacteria | 0.0992 | 22.69 |
| Total | 0.4373 | 100 |

Unique contributions measure the variance uniquely accounted for by a single variable; whereas common contributions indicate the variance that is common between two or more variables. The sum of all coefficients of each model corresponds to the total variance of TL explained by the model, which is the multiple $R^2$ (non-adjusted). Negative coefficients occur in presence of suppression or when some of the correlations among variables have opposite signs (see details in ref. 100).

Otherwise, the colony diameter has a null effect on *Pocillopora* spp. sTL (0.03% of the explained variance) and *Porites* TL variation (0.05% of the explained variance in hTL and 0.07% in sTL).

Table 1 displays the best models, while Supplementary Table 3 shows the results of the different models. For *Pocillopora* spp. hTLs, the selected model, based on the lowest Akaike Information Criterion (AIC), included island and Symbiodiniaceae and bacterial community composition in accordance with the variance partition combining a stronger effect of island, with a unique contribution of 48.1% of the total variance of hTL, with a smaller effect of both Symbiodiniaceae and bacterial community composition (Fig. 2). For *Pocillopora* spp. sTLs, the selected model included island and Symbiodiniaceae communities. In this model, the islands exhibit a unique contribution of 47.3% of the total variance of sTLs. For *Porites* spp. hTLs, the best model includes only the island. For *Porites* spp. sTLs, the selected model, based on AIC, included both island and Symbiodiniaceae communities, with a unique contribution of the islands of 62.5%. None of these models included colony diameter in agreement with the previous analyses that only detected a faint effect of this variable for *Pocillopora* spp.

In agreement with the lack of correlation between the coral TLs and the bacterial community composition, we found no significant Spearman correlations between the diversity index (Shannon) for bacterial communities[74] and coral TLs (*Pocillopora* spp. hTL and sTL: 0.06 and 0.05, respectively; *Porites* spp. hTL and sTL: 0.18 and 0.016, respectively).

Collectively, these results show that the island of origin is a prominent driver of TL variation both for *Pocillopora* spp. and *Porites* spp.

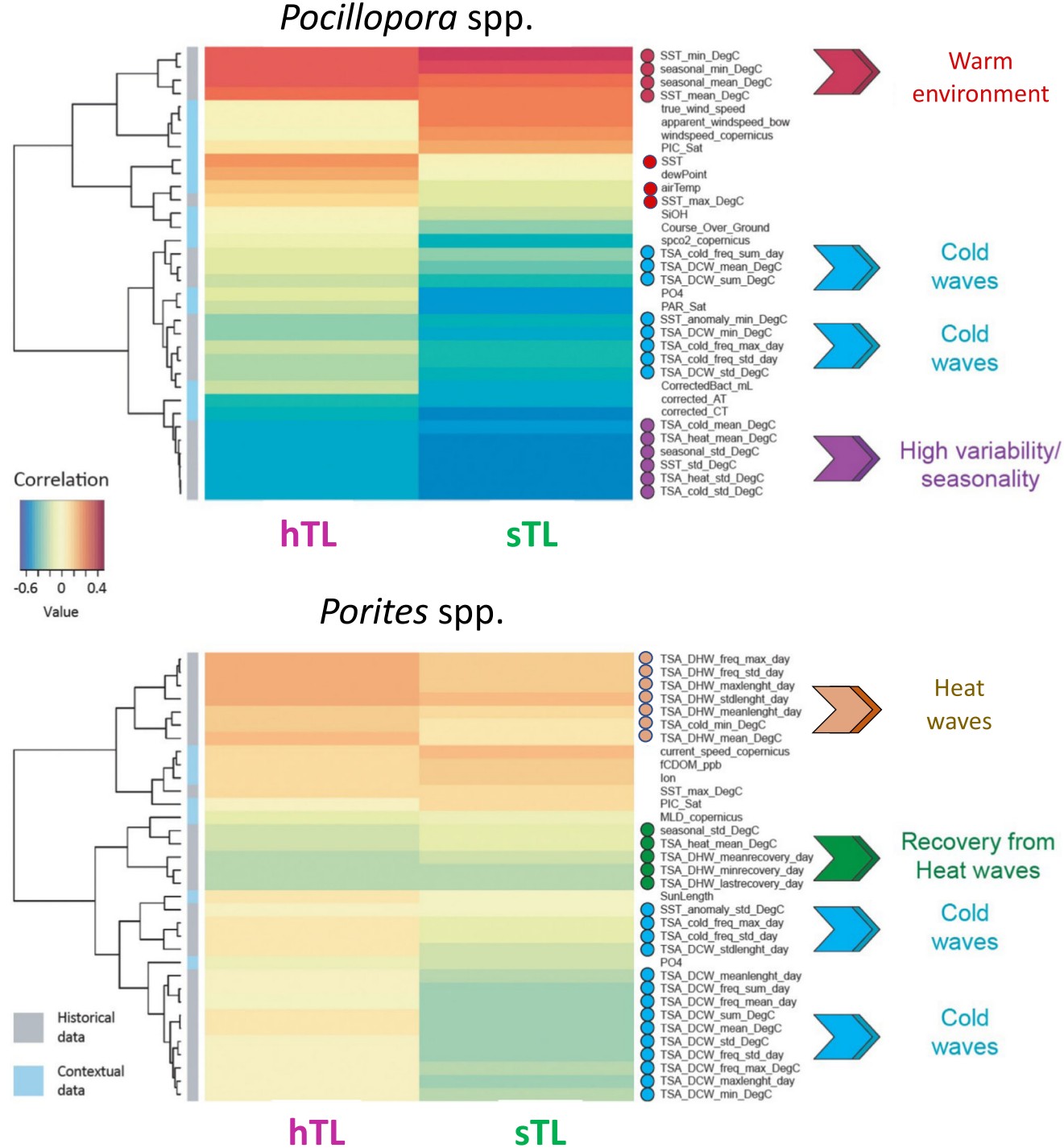

**Fig. 3 | Relationships between environmental variables and coral telomere DNA length variation.** Results of a two-dimensions sparse partial least squares (sPLS) regression of TL (hTL and sTL) and environmental variables (contextual at the time and sampling and historical as recorded from 2002 to the sampling date). Clustered image map of the two sPLS dimensions, displaying pairwise correlations between TL (bottom) and environmental variables (right). Red and blue indicate positive and negative correlations, respectively. Hierarchical clustering was performed within the mixOmics cim function based on the sPLS regression model. Source data are provided as a Source Data file.

colonies. This island effect can be associated with certain biogeographical characteristics. For *Pocillopora* spp. colonies, there was a global trend toward longer hTLs in samples collected from islands within or close to the Coral Triangle (South Palau Islands, Palau, Kimbe and Milney) (Fig. 1b and Supplementary Fig. 4), a bioregion of high marine biodiversity[75]. Short hTLs were found in corals from northern islands, such as Hawaii and Ogasawara, and from the southeast Rapa Nui Island. Less pronounced differences in TLs between islands were found for *Porites* spp. colonies except for Palau south Islands and Heron where the hTLs are markedly longer (Fig. 1b and Supplementary Fig. 4). Notably, the *Porites* spp. TLs from opposite sides of the Eastern Tropical Pacific[76] were not markedly different.

In summary, the island of origin explained 20–30% of the variance in hTL and sTL across the Pacific Ocean sampling sites, with distinct patterns in TLs variations between the two genera of reef-building corals. This unveils that telomere dynamics are different in two types

of reef-building corals sharing the same habitat and climate regime, suggesting genus-specific mechanisms of telomere maintenance.

## Respective contribution of host species and island on telomere DNA length variation

To determine whether TL variation is shaped by host species within each genus, we focused on samples from 11 islands for which host lineages are determined: three species for *Porites* spp. (K1–K3) and five species for *Pocillopora* spp. (SVD1–SVD5)[71,77].

Pairwise comparison of mean hTL between host species showed no significant differences for either *Pocillopora* spp. or *Porites* spp. (Supplementary Fig. 6a). This was confirmed by variance partitioning analysis (Supplementary Fig. 6b with the marginal and conditional $R^2$ values given in Supplementary Table 4) and linear regression models with a low or null effect of host species (Supplementary Table 5).

These analyses confirm the major driving effect of island of origin on the TL variation observed for the 32 islands (Fig. 2). For instance, the *Pocillopora* spp. species SVD5 had long (at Moorea) and short (at Rapa Nui) hTLs (Supplementary Fig. 6a, c). Nevertheless, some contribution of host species to TL cannot be ruled out, as suggested by the fact that SVD4 had similar hTLs among different islands (Coiba, Las Perlas, Ducie and Gambier Islands).

## Temperature history best explains the island effect on telomere DNA length

The above analyses revealed that island of origin is the major driver of coral TL, with Symbiodiniaceae and bacterial community composition, colony size, and host species having no or weak effects. This suggests that coral TL varies in association with environmental differences among islands. Following this, we analyzed hTL and sTL in reference to an extensive dataset of contextual (52) and historical (70) environmental variables determined for the 32 islands (for a complete set of variables queried, see Supplementary Tables 6 and ref. 59, the values of the environmental variables used for each colony are shown in Supplementary Data 1). These data were obtained from a suite of methods, including collected samples, automated onboard measurements, biogeochemical models, and satellite imagery[59]. To interpret the effect of local seasonal temperature fluctuations and past climatic events, we extracted high resolution (temporal and spatial) satellite sea surface temperature (SST) from the past 14–16 years (see complete methodology in ref. 59). From those, we extracted the mean seasonal variations and calculated the different heat waves (HW) and cold waves (CW) indices such as the degree heating/cooling weeks (DHW/DCW) and the Thermal Stress Anomaly (TSA) following the Coral Reef Temperature Anomaly Database (CoRTAD) methodology[78] and extracted their frequency, variability and recovery from last events. Of note, mean seasonal signal is defined as the mean-smoothed temperature variations over a mean year. Since we used historical indicators that are independent from the length of the time series, we are confident that they are only reflecting the average seasonal signals and the variability of deviations from this seasonal signal and are therefore valid to compare coral genera with different life-history.

To assess putative associations, we performed sparse partial least squares (sPLS) regression, because, firstly, it is well suited to handle environmental variables that are interrelated, and, secondly, it performs a selection to reduce the number of original variables. Heatmaps based on correlation coefficients (determined using the pairwise similarity matrix obtained from the results of the sPLS) obtained via a two-component sPLS model revealed that the best associations were between TL and variables linked to the Sea Surface Temperature (SST) history (considering approximately a 15-year time span, from 2002 to the sampling date) (Fig. 3). The plots of the sPLS model revealed a grouping of individual colonies by their island of origin, an effect more

pronounced for *Pocillopora* colonies (Supplementary Fig. 7). This indicates that the effects of island of origin can be explained, at least in part, by climate regime (Fig. 3).

## Distinct patterns of temperature-dependency between *Pocillopora* and *Porites* telomeres

For *Pocillopora* colonies, hTLs and sTLs were positively correlated with high mean seasonal SST (e.g., seasonal_mean and SST_mean) and winds and negatively correlated with indicators of high SST seasonality (e.g., seasonal_std and SST_std) together with various thermal stress anomaly (TSA) parameters related to cold water thermal stress (degree cooling weeks; DCW) (Fig. 3). Although DCW was a significant predictor, DHW was not. DHW is a common measure for coral bleaching susceptibility that indicates the amount of time exposed to above long-term MMMs (maximum monthly means)[79–81]. DCW is an analogous measure but related to cold events, that is much less commonly studied and considered[82].

For *Porites* colonies, the associations of TL with environmental factors were much weaker compared to *Pocillopora* (Fig. 3). Notably, hTLs and sTLs were not correlated with mean seasonal SST as were *Pocillopora* TLs but were positively correlated with presence/prevalence of past heating waves and negatively correlated with presence/prevalence of recovery between heating waves. It should be noted that *Porites* spp. sTLs were negatively correlated with cold wave variables including DCW, while heat waves had an opposite effect. A slight cold wave contribution to bleaching has been reported[83], suggesting that cold waves may stress the symbiont. However, the fact that the heat waves had an opposite effect on sTL in *Porites* spp. suggests that the symbiont reacts differently between cold and heat waves and/or when hosted in *Pocillopora* spp. or *Porites* spp[84].

Surprisingly, for *Pocillopora* spp., a coral genus known to be sensitive to climate change, we did not observe a clear relationship of TLs with heat wave parameters but rather with parameters reflecting seasonal thermal regimen. *Porites* spp., a coral genus known to be more resilient facing climate change, there is a slight effect of heat waves parameters. These results could be explained if *Pocillopora* spp. colonies, in contrast to *Porites* spp. colonies, exhibit a high rate of mortality after heat wave, the stress-resistant *Porites* spp. conserving some "TL sequels" of the past heat waves.

## Telomere gene expression correlates with hTL in *Pocillopora* spp

To investigate whether hTL variation is associated with particular patterns of host gene expression, we analyzed transcriptomic (RNA-sequencing) data from 55 *Pocillopora* spp. colonies of the subset of 11 islands used for host species assignment[85]. First, we selected genes whose expression significantly correlated with hTL (Spearman, $R > 0.3$, adjusted $p$-value $< 0.05$, 7289 out of 35,424 genes). Then, we evaluated the expression variance of these hTL-correlated genes attributable to islands of origin, host species, Symbiodiniaceae composition and hTLs (Fig. 4a, the marginal and conditional $R^2$ values are given in Supplementary Data 2). About half of the hTL-correlated genes exhibit an expression variance attributable to islands (Fig. 4a), in agreement with the major island effect on TL variation (Fig. 2).

All the genes whose expression variation is attributed to hTL (except *OSBPL8*) do not exhibit an expression variance attributable to the other variables (Supplementary Fig. 8a). Therefore, they can be considered as associated with hTL independently to island, Symbiodiniaceae composition and host species effects. The analysis of the Biological Process pathways of these genes obtained from the cnidarian annotations and from their human homolog annotations revealed redox-related gene ontology (GO) terms reflecting a positive correlation with two genes coding for heme-containing proteins (*PXDN* and *NGB*) and an anti-correlation with *OSBPL8* coding for an oxysterol binding protein (Supplementary Fig. 8b, c). The differential expression

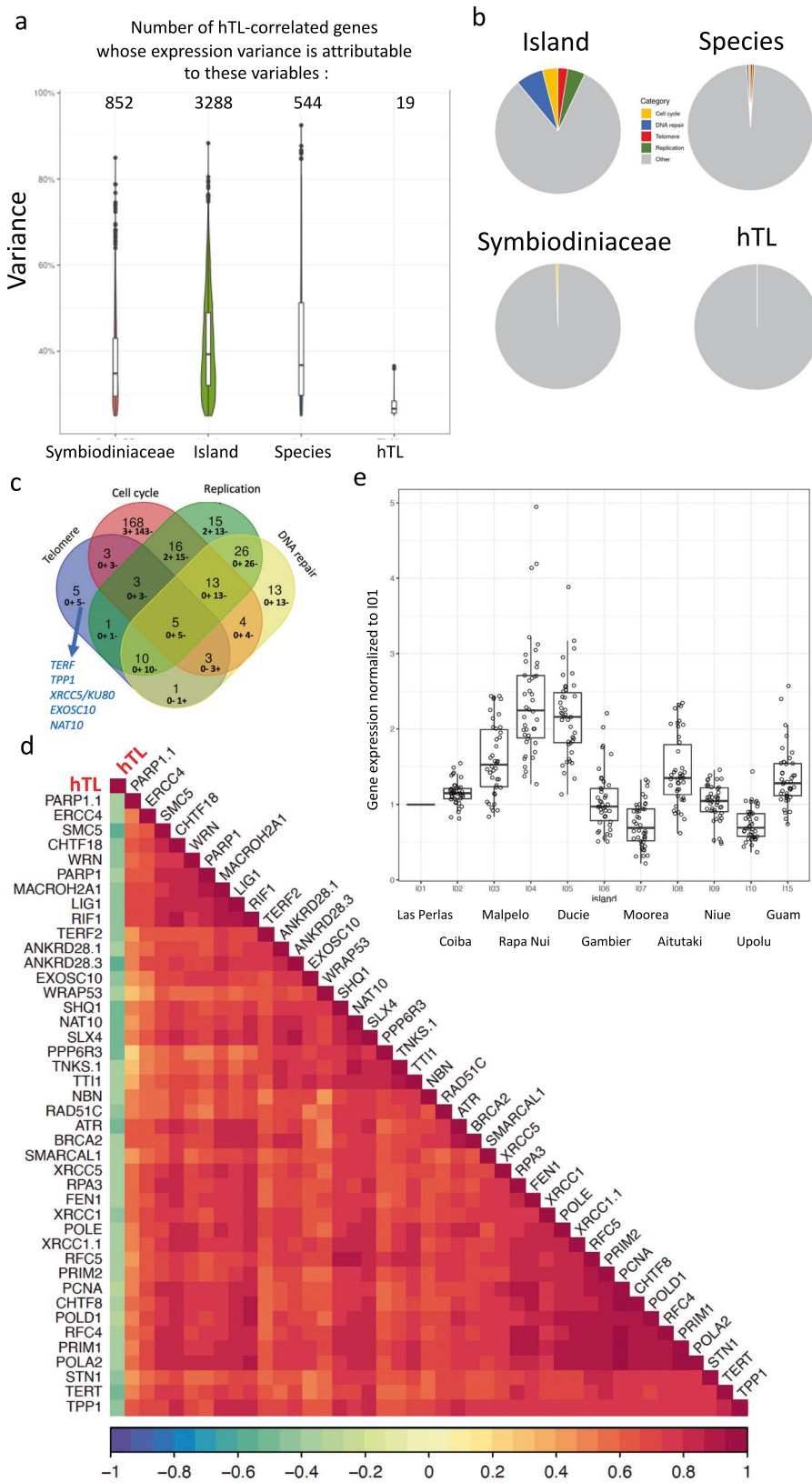

**a** Number of hTL-correlated genes whose expression variance is attributable to these variables:

**e** (island axis labels) Las Perlas, Coiba, Malpelo, Rapa Nui, Ducie, Gambier, Moorea, Aitutaki, Niue, Upolu, Guam

of *PXDN* and *NGB* among islands was mainly due to the higher expression levels for Rapa Nui. In contrast, *OSBPL8* was more highly expressed on Rapa Nui, Ducie and Gambier Islands (Supplementary Fig. 8c).

We also analyzed the functions of the hTL-correlated genes whose expression variation is associated with the islands, the Symbiodiniaceae composition and the host species. Notably, the functional analysis of the human homologs revealed a higher proportion of terms directly related to telomere, DNA repair, DNA replication, and cell cycle in the genes whose expression variation is associated with islands (Fig. 4b and Supplementary Data 3), a conclusion confirmed by a Reactome analysis (Supplementary Fig. 9).

**Fig. 4 | Negative correlation of coral telomere DNA length and expression of telomere genes in *Pocillopora* spp. a** Distribution of hTL-correlated genes for which more than 25% of the variation in their expression is explained by one of four predictor variables: the island of origin, the host species (shown as "species"), the Symbiodiniaceae communities and hTL. The number of genes in each distribution is indicated above each violin plot. **b** Pie-charts representing the proportion of significant (*p*-value < 0.05, Fisher's exact test) Biological Process pathways for human homologs. Five categories of pathways are defined: Biological Process pathways associated with the terms telomere, cell cycle, DNA replication and DNA repair determined from the genes whose expression variance is associated with island, Symbiodiniaceae composition, host species and hTL 'from panel **a**). The "Other" category gathers the remaining pathways. The full list of Biological Process pathways and the five categories that we defined are shown in Supplementary Data 3. **c** Venn diagram of the hTL-correlated genes found in the four categories of pathways defined in panel **b**: telomere, replication, cell cycle and DNA repair. Among them, the number of genes positively and negatively correlated with hTL

are indicated as: n+ m− (n genes positively correlated, m genes negatively regulated). The five hTL negatively correlated genes belonging only to the telomere category are indicated: TRF and TPP1 are two subunits of the telomere capping factor shelterin, XRCC5/KU80 is a DNA repair protein involved in telomere maintenance while NAT10 and EXOSC10are two proteins reported to be invoved in telomerase regulation. **d** Correlogram obtained with Spearman correlation of hTL and the expression of genes belonging to the telomere category (panels **b**, **c**); all significantly correlated between each other (*p* < 0.05). *p*-values were adjusted by the Benjamini−Hochberg method. **e** Expression by island of telomere genes selected in panel **c**. The expression (TPM) of each gene is normalized to the mean expression value of I01 (Las Perlas) (*n* = 103 samples). The boxplots of **a** and **e** are defined as follows: the lower and upper bounds of the box represent the first (Q1) and the third (Q3) quartile, respectively. The entire box represents the interquartile range (IQ). The median is represented as a line across the box. Whiskers extending from Q1 and Q3 are defined as 1.5xIQ. Source data are provided as a Source Data file.

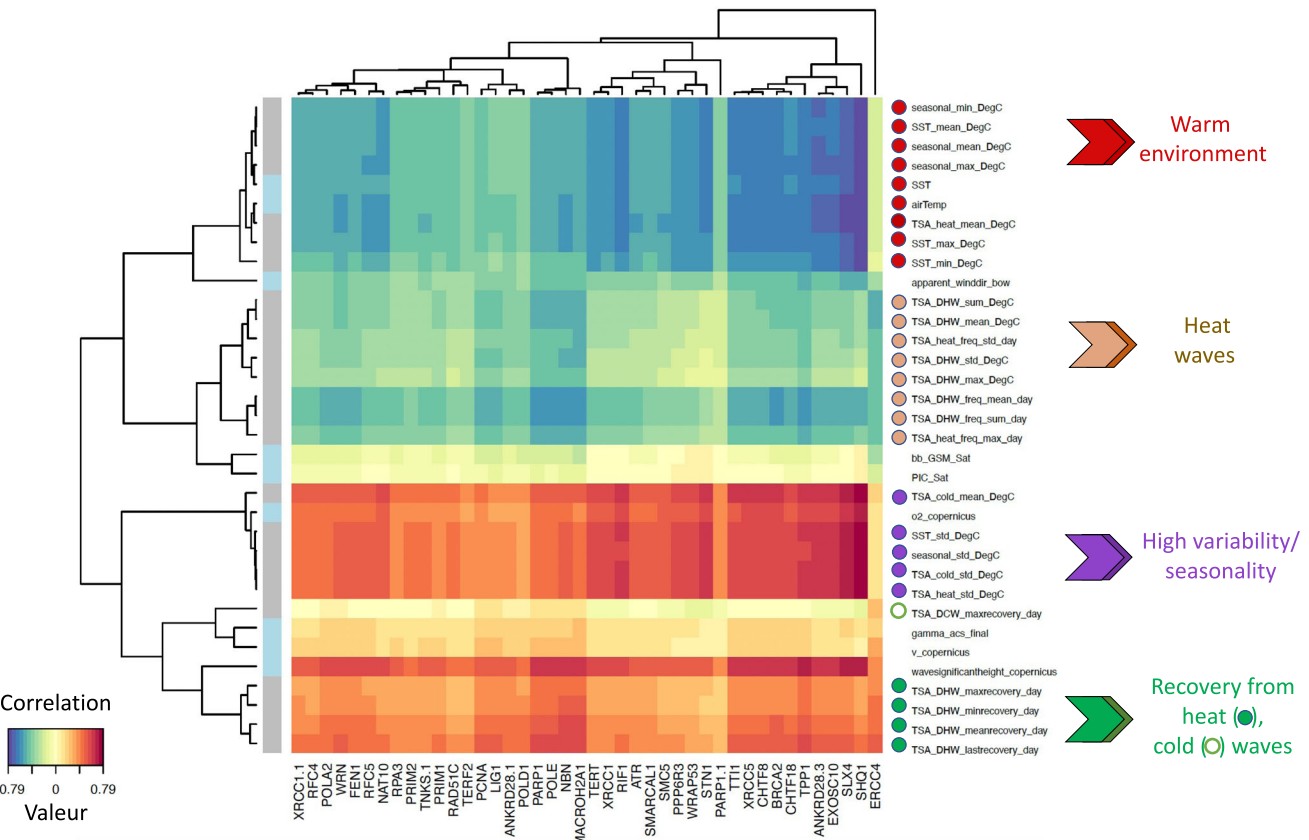

**Fig. 5 | Relationships between environmental variables and telomere gene expression.** Results of a two-dimensions sparse partial least squares (sPLS) regression of telomere gene expression (measured as TPM) and environmental variables (contextual at the time and sampling and historical as recorded from 2002 to the sampling date). Clustered image map of the two sPLS dimensions, displaying pairwise correlations between genes (bottom) and environmental variables (right). Red and blue indicate positive and negative correlations, respectively. Hierarchical clustering was performed within the mixOmics cim function based on the sPLS regression model. Source data are provided as a Source Data file.

Noteworthy, these results were obtained when we analyzed the function of the human homologs not of the cnidarian genes, most likely revealing limited information related to these functions in the cnidarian annotations. This, together with the fact that hTL is mainly determined by island effects, suggest that the expression of these genes play an important role in hTL regulation. Indeed, among the 42 genes associated with the terms "telomere or telomerase", some are known to have specific and key functions for telomere replication, notably those encoding for telomerase subunits (the catalytic subunit TERT and the accessory factors NAT10, WRAP53/TCAB and SHQ1), for the STN1 subunit of the CST complex required for telomere replication

resolution and for telomere protection (in particular the two shelterin subunits TRF and TPP1) (Fig. 4c, d). Impressively enough, the expression of all these genes is negatively correlated with hTL while positively correlated between each other (Fig. 4c, d). Moreover, their expression levels exhibited similar patterns among islands, including a higher expression level in Rapa Nui and Ducie (Fig. 4e). These results reveal a global transcriptional telomere signature negatively correlated with hTL. This signature can be extended with most genes involved in cell division, DNA repair, DNA replication, and cell cycle since they also exhibit a negative correlation with hTL and a positive correlation between them (Supplementary Fig. 10). In accordance with this trend,

these genes follow the same pattern of expression among islands as those with telomere functions (Supplementary Fig. 11). Taken together, these results indicate a coordination between the expression of genes involved in telomere maintenance and those related to cell division, a result in accordance with the tight connections existing between telomere, replication, DNA repair and cell division.

These results led us to explore whether the expression of the telomere genes is linked to environmental conditions. We explored this connection by performing a sPLS regression. The heatmap based on correlation coefficients obtained via a two-component sPLS model revealed a high expression linked to indicators of high SST seasonal variability and low expression in a warm environment (Fig. 5). This analysis also revealed lower correlation patterns: a positive one with recovery from heat/cold wave indicators and a negative one with heat waves indicators. Pairwise correlations confirmed these results (Supplementary Fig. 12). The plots of the sPLS model revealed a grouping of individual colonies by their island of origin (Supplementary Fig. 13), suggesting that the telomere gene expression pattern is connected, at least in part, to island effects. This is reminiscent of the TL variation determined by the island of origin (Fig. 2 and Supplementary Fig. 7). These results agree with the negative correlation between telomere gene expression and TL (Fig. 4).

Both the temperatures at the time of sampling (contextual indicators: SST, airTemp) and the thermal regimes (historical indicators: SST-mean, SST, min, SST-max, seasonal-mean, seasonal-min and seasonal-max) were negatively correlated with the expression of telomere genes (Fig. 5), in agreement with the sPLS heatmap of Fig. 3 showing a positive correlation between contextual and historical thermal indicators and hTL and Fig. 4 showing a negative correlation between hTL and telomere gene expression. Therefore, the expression of telomere genes varies in the same direction whether it is based on historical or contextual thermal indicators, suggesting that the expression of telomere genes at the time of sampling is coupled to the current TL measures and reflects telomere gene expression pattern over time.

## Discussion

At the level of the entire Pacific basin, natural variation in reef-building coral TL was best explained by differential patterns of thermal climates between coral genera with distinct life-history traits. To the best of our knowledge, no similar integrated, holistic study comparing telomere biology between two related organisms at such a high resolution and large ecological scale has been attempted before, making this study unique. Moreover, this represents the first field work on the thermal effects on TL variations in aquatic invertebrates[13]. Specifically, TL was largely determined by seasonal temperature history in the stress-sensitive, short-lived *Pocillopora* species, whereas it was resilient under varying seasonal temperatures in stress-resistant, long-lived *Porites* species. These pattern differences (*Pocillopora* spp. *vs.* *Porites* spp.) hold true beyond the species level and are applicable to the genera *Porites* and *Pocillopora* at large (see Supplementary Fig. 6b). Overall, these results unveil that the mechanisms regulating TL in response to environmental conditions exhibit differences between coral genera with different life-history traits.

As TL regulation is intimately coupled to DNA replication and regeneration[4], the difference of TL variation linked to thermal regime between *Pocillopora* spp. and *Porites* spp. might be associated with growth rates. Since a warmer temperature can increase coral growth[86,87] and, in starfish, arm regeneration can lead to TL elongation[88], warmer seasonal conditions might elongate TL due to an increased growth rate. However, the reverse was observed, i.e., a trend toward hTL shortening in the apex of *Pocillopora* spp. branches where coral tissues expand (Supplementary Fig. 1) and in larger *Pocillopora* spp. colonies (Supplementary Fig. 5a), indicating that coral growth is rather associated with TL shortening than lengthening in *Pocillopora* spp.. This is further supported by the transcriptional analyses of

*Pocillopora* spp. colonies in 11 islands, showing a negative correlation between hTL and a large signature of genes involved DNA replication, DNA repair and cell cycle (Supplementary Fig. 10), suggesting a TL shortening linked to the growth of *Pocillopora* spp. tissues. Thus, growth rate cannot merely explain the longer telomeres of *Pocillopora* spp. in warmer islands within or close to the Coral Triangle and therefore the differences between *Pocillopora* spp. and *Porites* spp. TL regulations. To explain the long TLs in warm islands, one can speculate that these corals live close to their optimum temperature, lowering stress-induced oxidative damage. Analyzing the co-variation of TL and temperature performances in corals might be a useful approach to explore this hypothesis[13]. It might also be that under protective and unchallenged conditions, the cost of maintaining long telomeres is better supported. If this long TL phenotype might have no other beneficial effect than ensuring chromosome end stability, the short telomeres encountered under stressful conditions may have specific adaptive values, for instance by being more prone to signal survival pathways and cope with energetic trade-offs as was previously proposed[26,47,48,89]. Nevertheless, this cross-sectional and field study cannot fully disentangle whether the association between TL and environmental conditions is a cost paid by individuals, or a result of the adaptation to different thermal regimes. To address the question of the adaptive value of the coral TL variation as a function of environment and island of origin, future studies using population telomere data to predict coral species habitat suitability through species distribution models (SDMs) may be helpful[21].

Our results lead us to revisit the question of whether the way TL is regulated by environmental conditions is a corollary or a driver of life-history traits. If the coral telomeres, like in a wide range of organism, including ectotherms, play a role in stress-response, health, and aging, one can hypothesize that the telomere-environment relationships of *Pocillopora* spp. and *Porites* spp. contribute to their differences in stress-resistance and longevity properties. For example, the development of efficient TL resilience mechanisms for somatic maintenance in response to environmental changes might increase longevity, as suggested by the TL resilience in *Porites* spp.. The sensitivity of *Pocillopora* spp. TL to temperature might render this coral less stress-resistant and short-lived than *Porites* spp..

Our results suggest that the mechanisms of TL regulation in response to environmental changes evolved differently between *Pocillopora* spp. and *Porites* spp. The comparative analysis of the genetic structures between *Pocillopora* spp. and *Porites* spp. suggests that the polymorphism associated with the environment is more conserved and possibly ancestral among *Porites* spp. than among *Pocillopora* spp[71]. Therefore, the apparition of the genetic determinants of the TL regulation resilience in *Porites* spp. might be linked to genus-specific life-history traits, e.g., stress resilience. Noteworthy, genomic analyses of the *Porites* genome revealed an intriguing repeated sequence containing a palindromic telomeric sequence uniquely found in the *Porites* genomes[90]. Whether these sequences are involved in the resilience of *Porites* telomeres to environmental changes is an interesting hypothesis to investigate.

A first insight on the mechanisms regulating telomeres in *Pocillopora* spp. can be derived from a transcriptomic analysis of corals around 11 islands. Although the level of correlation between gene expression and TL was generally weak ($r > 0.3$, $p < 0.05$), it is impressive that the expression levels of large numbers of key telomere maintenance genes (including those encoding the telomerase catalytic subunit TERT and the telomere capping proteins TRF and TPP1), were negatively correlated with TL. Moreover, the expression of telomere genes is coupled to the thermal climate indicators (both contextual and historical) in an inverse pattern than TL variation. Altogether, these results indicate that the expression of telomere genes and hTL are coupled to the thermal climates of the islands where the *Pocillopora* spp. colonies were collected. Noteworthy, shorter TLs combined

with higher expression levels of telomere and cell division genes is particularly apparent in the Rapa Nui Island, where the *Porites* spp. TLS are also short (Fig. 4d, Supplementary Fig. 6c and 11). This suggests distinct biogeographic and seasonal features in this island that link growth to short TLs, for instance, the shorter sexual reproduction period of corals at Rapa Nui[91]. The odd association in *Pocillopora* spp. between short TLs and increased expression of the telomerase genes could be explained by the concomitant high level of the gene expressing the telomere protein TRF known to behave as a negative regulator of telomere extension by telomerase[92,93]. Such a combination of short telomeres with high levels of *TERT* and/or *TERF* gene expression is not unprecedented since shorter TLs with high telomerase activity are associated in humans with high allostatic load[94] while some human cancer cells combine shorter TLs, elevated telomerase activity and high TRF2 expression[95]. Overall, these results suggest a transcriptional response to environmental changes in *Pocillopora* spp. that reinforces the protection of short telomeres during coral growth.

The biological significance of these findings may extend beyond corals. Notably, the hTL differences between colonies (3–10 kb for *Pocillopora* spp. and 3–14 kb for *Porites* spp.) were within the range of TL variation observed in human and zebrafish ageing (4–12 kb)[96,97]. Therefore, even though the TLs of adult coral colonies and human/ zebrafish individuals differ in their dependence on chronological age, they oscillate between similar limits, suggesting that, as in these two vertebrates, variations in coral TL may have biological significance regarding health, resilience, and longevity. Finally, appropriate mechanisms linking telomere response to the environment could be particularly important for sessile organisms, like plants, to counteract the deleterious effects of seasonal variations[6,13].

In summary, natural variation in reef-building coral TL (in both the coral host and the algal endosymbiont) was best explained by thermal climate but not by colony size (used as a proxy of age) and exhibit different patterns of response to climate regimes between coral genera with different life-history traits. These results suggest that the way telomeres are regulated by the environment is coupled to life-history traits. They lay the groundwork for new mechanistic and comparative studies to explore the causal or correlative relationships between TL regulation by environmental factors and life-history trait evolution. Our findings also suggest that climate change may impact telomere homeostasis in a coral genus-specific manner, a determinant of coral health and biodiversity to consider in future reef restoration interventions. Additionally, they allow us to think about how to target telomere maintenance pathways for preventing and treating the adverse effects of heat waves on health and ageing in humans.

## Methods

Our research complies with all relevant ethical, institutional and international regulations applying to field study of coral samples.

### Sampling

Samples were collected from coral colonies around 32 Pacific Ocean islands. For all these islands, there were at least three sampling reef sites, representing 99 reef sites in total[58]. For preparing the $T_2AG_3$ probe, we used the following primers: F: GGGTTAGGGTTAGGGT-TAGGGAAA and_R: TTTCCC TAACCCTAA. For preparing the $T_3AG_3$ probe we used the following primers: F: GGGTTTAGGG TTTAGGG TTTAGGGAAA and R: TTTCCCTAAACCCTAAA. The sampled colonies of *Pocillopora* spp. were targeted based on their resemblance to *Pocillopora meandrina* and those of *Porites* spp. to *Porites lobata*. The genotyping of colonies of a subset of 11 islands confirmed that the sampled colonies are phylogenetically related to the targeted species[71,77]. Samples were flash-frozen in liquid nitrogen on board and kept stored at −20 °C before DNA extraction and TL analysis.

For the additional sampling with size classification (Supplementary Fig. 5c-f), *Porites* spp. were collected from sites in Palau ($n = 12$) and

Clipperton Island ($n = 14$), while *Pocillopora* spp. samples were collected from sites on Clipperton Island ($n = 13$).The colonies were classified according to size as follows: (S1, diameter <15 cm; S2, diameter = 15–24 cm; S3, diameter = 25–50 cm; and S4, diameter > 75 cm).

### Telomere restriction fragment (TRF) assay

DNA extraction, probe preparation, and Southern blotting procedures are described elsewhere[57]. To control for the telomeric origin of the observed Southern signal we performed a Bal31 experiment by adding 2 U of the exonuclease Bal31 (New England Biolabs, M0213S) to 5 μg of sample DNA and incubated the sample at 30 °C for 0, 5, 10, 15, or 30 min. Next, the samples were inactivated for 10 min at 65 °C with EGTA (30 mM). DNA was precipitated and digested overnight for telomere restriction fragment assay (Supplementary Fig. 1a).

Samples with significant DNA degradation were excluded from the analysis. In addition, two blinded observers assessed signal quality of the Southern blot by reporting lanes without signal or with a poor quality one; these were removed from the analysis.

### Image analysis

Telomere Restriction Fragment (TRF) images signal was extracted using ImageQuant (GE Healthcare) 1D gel analysis mode and manual lane creation. Ladder signal intensities were extracted from Ethidium bromide gel images, setting the lane upper limit at the gel wells bottom. Pixel position of ladder peaks were manually reported. Host and symbiont telomere signals were extracted from the phospho screen images, exposed to radioactive labeled membrane. Efficiency of stripping step between the two probes hybridization was assessed imaging phosphoscreens overnight exposed to stripped membranes[57].

### Telomere length measurements

We used two types of ladders, the SmartLadder (Eurogenetec) for (10 kb–0.2 kb) and the QuickLoad 1 kb Extend (NEB) for (48.5 kb–0.5 kb). They were loaded in two successive lanes at the left and right ends of the gel. In some gels where the number of samples allowed it, the two ladders were also loaded in a middle. Single lane intensity files were fused in R. Ladder peaks exact position was extracted in R, searching for the maximum intensity in a 10-pixel perimeter around the manually reported peaks. Since higher molecular weights are migrating faster while lower ones are migrating more slowly, we used two different linear equations to extrapolate the size more accurately over the entire gel length. Fitted linear model coefficients (a,b) of log2 ladder size (kb) against peak pixel position were calculated with the high molecular weight ladders sizes (48.5 kb, 20 kb and 15 kb) and with the low molecular weight ladder sizes (10 kb, 8 kb, 6 kb, 5 kb, 4 kb, 3 kb, 2.5 kb, 2 kb) using the lm function of "stats" R package. The position of the switch between the low and the high molecular weights equation depended on the continuity of the scale and was slightly different between gels but always fell into an 8kb-10kb range. Coefficients were used to transform samples intensity scale from pixel to base pair (bp) in Excel using the high molecular weight coefficient for the upper part and the low molecular coefficient for the lower one. Depending on their position on the membrane samples were divided in left, right and middle to be scaled to the closest ladder. Intensity signals were imported in R using the read_excel from "readxl" package and merged using "rowr" package, background correction was automatically computed to level the signal by subtracting to each position the minimal intensity with "reshape2" package. Intensity was normalized by the size to avoid probe number hybridization bias (using the equation described in ref. 98), intensity below 2 kb were discarded to avoid genomic noise, interstitial telomeric sequence noise and normalization bias (<1 kb). Besides the mean TLs, we calculated a suite of parameters that describe the shape of the TL distributions to identify any skewness toward short or long TLs: median

(med), 1st quartile TL limit (Q1), 3rd quartile TL limit (Q3) and inter-quartile distance (IQ). For each genus, there was a significant correlation among these parameters both for hTLs and sTLs (Supplementary Fig. 2). The results of the measurements for all colonies (hTLs and sTLs) are shown in Supplementary Fig. 3. The mean TLs estimated from independent TRF experiments are highly correlated (Pearson's test, $R = 0.7233599$; $p < 2.2e^{-16}$; $n = 226$ for hTL, and $R = 0,8953802$; $p < 2.2e^{-16}$; n = 117; Supplementary Fig. 3c).

## TL mapping
GPS locations were retrieved from the sampling site, and the scale was converted from (−180°/180°) to (0°/360°). Maps were generated using the *map_data* function from the "ggplot2" R package. Pie-charts were generated with "ggplot2" R package using the mean TL at each site. To prevent overlaps of pie-charts between islands, locations were slightly shifted to the top, right, left or bottom depending on the island.

## Correlation analysis
Correlations between TL measurements and colony size were computed using the *ggpairs* function in the "GGally" R package, with the *method = "spearman"* option. Correlations between gene expression levels were computed using the *rcorr* function in the "Hmisc" R package, with the *type = "spearman"* option. Correlograms were generated using the *corrplot* function in the "corrplot" R package. *p*-values were adjusted by the Benjamini–Hochberg method using the *p.adjust* function in the "stats" R package.

## Variance partitioning analysis
Variance partitioning analysis was performed using the "variancePartition" R package[99]. The marginal and conditional $R^2$ values were computed for each model using the *r2* function from the "performance" R package.

Data for microbial communities are based on a clustering approach that groups similar Symbiodiniaceae *(ITS2)* and bacterial (16S rRNA) communities into clusters. Symbiodiniaceae communities were clustered into 47 groups employing partitioning around medoids using Bray-Curtis dissimilarity-generated distances based on square root transformed ITS2 sequence counts (post-MED table[73]). Microbiomes were separated into 7 and 8 clusters for *Pocillopora* and *Porites*, respectively[58].

To analyze hTL/sTL variance among the 32 islands, TL values (i.e., mean, median, Q1, Q3, and IQ) and colony diameter mean were standardized with the *scale* function from "base" R package. The colony diameter mean was included as fixed effect, island of origin and Symbiodiniaceae and bacterial community compositions were random effects.

In the analysis of gene expression, island of origin, host species, and Symbiodiniaceae composition were included as random effects, and hTL was a fixed effect. Normalized counts (TPM) were used for the analysis. Genes for which >25% of the variance in expression was explained by one of the four variables were subjected to further analysis.

To analyze hTL/sTL variance among a subset of 11 islands, host species and island of origin were included as random effects.

## Regression models
All models were generated using the *lm* function from the "stats" R package. For each species and TL, all the possible models were generated, and the best model was selected according to its lowest AIC value. Then a commonality analysis was performed to determine the unique and common contribution of the different variables to the TL variation[100]. The contribution coefficients were calculated using the function *commonalityCoefficients* from the "yhat" R package.

## Sparse partial least squares analysis
sPLS analysis was used to determine historical and contextual environmental variables associated with hTL/sTL, because it can handle multicollinearity and noisy data. The sPLS was performed using the *spls* function in the "mixOmics" R package[101]. We set the number of components to 2. All the telomeric variables were retained for both components (*keepY = 2*), as well as 15% of the environmental variables (*keepX*). Clustered image maps (CIMs) were used to visualize the hierarchical clustering and correlations between blocks of variables. CIMs were generated using the *cim* function in the "mixOmics" R package. Finally, individual plots were generated using the *plotIndiv* function (*rep.space = "XY-variate"*).

## RNA extraction, sequencing, and gene expression levels of Pocillopora
Coral fragments were processed to extract and isolate RNAs as described by Belser et al.[61]. Briefly, DNA and RNA were extracted simultaneously from these aliquots using Quick-DNA/RNA Kits (Zymo Research, CA, USA). Eluted RNA was stored at −80 °C prior to library construction. Poly-A + RNA libraries were prepared following the Tru-Seq Stranded mRNA sample preparation protocol (Illumina, San Diego, CA, USA). RNA libraries were sequenced using 151 bp paired-end read chemistry on a NovaSeq or HiSeq4000 Illumina sequencer (Illumina, San Diego, CA, USA). Short transcriptomic reads (<30 bp length), low-quality nucleotides (*Q*-score < 20) and adaptor/primer sequences were removed with an in-house script based on Fastx-Toolkit software (https://github.com/institut-de-genomique/fastxtend), as well as read pairs that mapped to the Enterobacteria phage PhiX174 genome (GenBank: NC_001422.1). Read pairs that mapped to ribosomal sequences were removed using SortMeRNA v2.186. After these filtering, we obtained between 39 M and 104 M metatranscriptomic reads. These sequences were separately aligned to predicted coding sequences (CDS) of the *Pocillopora meandrina* coral host genome[90]. Read counts were normalized as transcript per million (TPM). Tables of raw and normalized counts are available on *Zenodo*[102].

## Human homologs
*Pocillopora* proteins were predicted from the *Pocillopora cf. effusa* genome as described in ref. 90. Briefly, CDSs were predicted using the mapping of proteins from 18 Cnidarian species and *Pocillopora cf. effusa* transcripts with Gmove tool[103]. Based on putative exons and introns from the alignments, Gmove searched for open reading frames (ORFs) consistent with the protein evidence. Finally, putative transposable elements (TEs) were removed as detailed in ref. 90. In addition, human homologs of *Pocillopora cf. effusa* proteins were obtained using a BLASTp[104] search of the UniProtKB/Swiss-Prot database, restricted to *Homo sapiens*. An *e*-value threshold of 1e-5 was set to filter the results. For each *Pocillopora cf. effusa* protein, the best hit based on *e*-value and bit score was selected.

## Gene ontology (GO) terms enrichment analysis
For the human analysis, all GO terms, and their associated genes were gathered from the Ensembl BioMart tool[105]. For the cnidarian analysis, GO terms were predicted from the protein sequences using the InterPro database[106] for *Pocillopora* and *Porites* separately. GO terms enrichment analysis was performed using the runTest function from the "topGO" R package[107] with a Fisher's exact test ("fisher" option). The threshold for the *p*-value was set at 0.05 and no correction for multiple testing was carried out as recommended by the authors.

## Reporting summary
Further information on research design is available in the Nature Portfolio Reporting Summary linked to this article.

## Data availability

The Telomere DNA length (TL) data generated in this study have been deposited in the *Zenodo* database[64]. The other variables used in this work are available as follows: environmental parameters (historical and contextual)[108]; RNAseq[102]; symbiodinaceae (ITS2) community[109]; bacterial (16 S rRNA) community[110] colony size[:71] and species delimitation[71,77]. Source data are provided with this paper. The genomic data used in this study are available in the European Nucleotide Archive (ENA) under the umbrella project PRJEB47249 with the ITS2 amplicon, metatranscriptomic, and metagenomic reads stored within projects PRJEB52458, PRJEB52301, and PRJEB52368, respectively. The sequencing data, genome assemblies and gene predictions of Porites lobata, Porites evermanni and porites. cf effusa are available in ENA under the following project [https://www.ebi.ac.uk/ena/browser/view/PRJEB51539], and at the following website http://www.genoscope.cns.fr/corals. Source data are provided with this paper.

## Code availability

R (v3.6.3) was used for statistical analysis and to generate plots (used packages are in brackets): data curation (readxl, stringi, stringr, reshape2, rowr, devtools and tidyverse),boxplots (ggplot2, RColorBrewer and plotly), correlograms (corrplot and GGally), functional analysis (topGO, Reactome and ggplot2), variance partition (variancePartition and ggplot2), sPLS (mixOmics), regression models and statistical analysis(stats, yhat and multcompView), maps (marmap and ggplot2). No custom code was generated.

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

## Acknowledgements

Special thanks to the *Tara* Ocean Foundation, the R/V *Tara* crew and the *Tara* Pacific Expedition Participants (*Tara* Pacific Consortium. (2020). *Tara* Pacific Expedition Participants. *Zenodo*.https://doi.org/10.5281/zenodo.3777760). We thank Jean-François Lemaître and Florentin Remot for advice on biostatistical analyses and critical reading of the manuscript. We are keen to thank the commitment of the following institutions for their financial and scientific support that made this unique *Tara* Pacific Expedition possible: CNRS, PSL, CSM, EPHE, Genoscope, CEA, Inserm, Université Côte d'Azur, ANR, agnès b., UNESCO-IOC, the Veolia Foundation, the Prince Albert II de Monaco Foundation, Région Bretagne, Billerudkorsnas, AmerisourceBergen Company, Lorient Agglomération, Oceans by Disney, L'Oréal, Biotherm, France Collectivités, Fonds Français pour l'Environnement Mondial (FFEM), Etienne Bourgeois, and the *Tara* Ocean Foundation teams. *Tara* Pacific would not exist without the continuous support of the participating institutes. The work was supported by the ANR CORALGENE, FRANCE GENOMIQUE (ANR-10-INBS-09). "Investments for the Future" programs LABEX SIGNALIFE ANR-11-LABX-0028, IDEX UCAJedi ANR-15-IDEX-01, the Diversity of Biological Mechanisms (DBM) CNRS grant and the AGEMED cross-cutting program of Inserm. This is publication number #00 of the *Tara* Pacific Consortium.

## Author contributions

A.R. designed the experiments, extracted the coral DNAs, co-executed and analyzed the TRF experiments, supervised students, performed data analysis, and wrote the manuscript; M.P. designed and executed the bioinformatic and biostatistical analyses and wrote the manuscript; N.D. co-executed the TRF experiments; Q.C., E.A., and J.L.H. helped for RNAseq analyses; B.P. and R.Mc.M. helped for biostatistical analyses; A.O., L.D.M., C.L., J.T., L.W., M.J.G.P., and D.Z. helped for coral DNA extraction and TRF experiments; G.B. and F.B. helped for environmental and historical data analyses; J.M.A. and D.F. helped for genetic data analyses; P.W., P.F., E. Boissin, E. Boss, R.D., J.P., G.S., H.J.R., B.C.H.,

P.E.G., S.A., S.S., R.M., D.A., H.J.R., D.P.G., P.W., E.R., S. Pesant, C.d.V., B.B., C.B., E.D., M.F., S. Reynaud, O.T., R.V.T., D.A., S. Planes, and C.R.V. shared unpublished data, provided useful comments and manuscript edition; R.T. led the Tara-Pacific expedition with SPl and DA as scientific directors; E. Boissin, G.I., G.B., A.P., and S. Romac conceived, planned, and performed onboard sampling. C.M. facilitated sample collection and obtained sampling permits; E.G. conceived the project, coordinated the work, supervised and performed the analyses, secured funding and wrote the manuscript. All authors contributed to the interpretation of the results and edited the manuscript.

## Competing interests

The authors declare no competing interests.

## Additional information

Alice Rouan [1,2,27] ✉, Melanie Pousse[1,2,3,27], Nadir Djerbi[1,2,3,27], Barbara Porro[1,2,3], Guillaume Bourdin [4], Quentin Carradec [5,6], Benjamin CC. Hume[7], Julie Poulain[5,6], Julie Lê-Hoang[5,6], Eric Armstrong[5,6], Sylvain Agostini [8], Guillem Salazar[9], Hans-Joachim Ruscheweyh [9], Jean-Marc Aury [5,6], David A. Paz-García[10], Ryan McMinds [1,11,12], Marie-Josèphe Giraud-Panis[1,2,3], Romane Deshuraud[1,2,3], Alexandre Ottaviani [1,2,3], Lycia Die Morini[1], Camille Leone[1], Lia Wurzer[1], Jessica Tran[1], Didier Zoccola [2,13], Alexis Pey[1,2,3], Clémentine Moulin[6,14], Emilie Boissin [15], Guillaume Iwankow[15], Sarah Romac [16], Colomban de Vargas[6,16], Bernard Banaigs[15], Emmanuel Boss [4], Chris Bowler [6,17], Eric Douville [18], Michel Flores[19], Stéphanie Reynaud[2,13], Olivier P. Thomas[20], Romain Troublé [6,14], Rebecca Vega Thurber[21], Serge Planes[6,15], Denis Allemand [2,13], Stephane Pesant[22], Pierre E. Galand [6,23], Patrick Wincker[5,6], Shinichi Sunagawa [9], Eric Röttinger [1,2,3], Paola Furla [1,2,3], Christian R. Voolstra [7], Didier Forcioli [1,2,3], Fabien Lombard [6,24,25] & Eric Gilson [1,2,3,26] ✉

¹Université Côte d'Azur-CNRS—Inserm—Institute for Research on Cancer and Ageing, Nice (IRCAN), Medical School, Nice, France. ²Laboratoire International Associé Université Côte d'Azur—Centre Scientifique de Monaco (LIA ROPSE), Monaco, Nice, France. ³Institut Fédératif de Recherche—Ressources Marines (IFR MARRES), Université Côte d'Azur, Nice, France. ⁴School of Marine Sciences, University of Maine, Orono, ME, USA. ⁵Génomique Métabolique, Genoscope, Institut François Jacob, CEA, CNRS, Univ Evry, Université Paris-Saclay, 91057 Evry, France. ⁶Research Federation for the Study of Global Ocean Systems Ecology and Evolution, FR2022/Tara Oceans GO-SEE, 75016 Paris, France. ⁷Department of Biology, University of Konstanz, Konstanz, Germany. ⁸Shimoda Marine Research Center, University of Tsukuba, Shimoda, Japan. ⁹Department of Biology, Institute of Microbiology and Swiss Institute of Bioinformatics, ETH Zurich, 8092 Zurich, Switzerland. ¹⁰Centro de Investigaciones Biológicas del Noroeste (CIBNOR), Av. IPN 195, La Paz, Baja California Sur, 23096 La Paz, México. ¹¹University of South Florida Center for Global Health and Infectious Diseases Research, Tampa, FL, USA. ¹²Maison de la Modélisation, de la Simulation et des Interactions (MSI),, Université Côte d'Azur, Nice, France. ¹³Centre Scientifique de Monaco, Principality of Monaco, Monaco, Monaco. ¹⁴Tara Ocean Foundation, 8 rue de Prague, 75012 Paris, France. ¹⁵Laboratoire d'Excellence "CORAIL," PSL Research University: EPHE-UPVD-CNRS, USR 3278 CRIOBE, Université de Perpignan, Perpignan Cedex, France. ¹⁶Sorbonne Université, CNRS, Station Biologique de Roscoff, AD2M, UMR 7144, ECOMAP, Roscoff, France. ¹⁷Institut de Biologie de l'Ecole Normale Supérieure (IBENS), Ecole normale supérieure, CNRS, INSERM, Université PSL, 75005 Paris, France. ¹⁸Laboratoire des Sciences du Climat et de l'Environnement, LSCE/IPSL, CEA-CNRS-UVSQ, Université Paris-Saclay, 91191 Gif-sur-Yvette, France. ¹⁹Weizmann Institute of Science, Department of Earth, and Planetary Sciences, 76100 Rehovot, Israel. ²⁰School of Biological and Chemical Sciences, Ryan Institute, University of Galway, University Road, H91TK33 Galway, Ireland. ²¹Oregon State University, Department of Microbiology, 220 Nash Hall, Corvallis, OR 97331, USA. ²²European Bioinformatics Institute, Wellcome Genome Campus, European Molecular Biology Laboratory, Wellcome Genome Campus, Cambridge CB10 1SD, UK, UK. ²³Sorbonne Université, CNRS, Laboratoire d'Ecogéochimie des Environnements Benthiques (LECOB), Observatoire Océanologique de Banyuls, Banyuls-sur-Mer, France. ²⁴Sorbonne Université, Institut de la Mer de Villefranche sur mer, Laboratoire d'Océanographie de Villefranche, Villefranche-sur-Mer, France. ²⁵Institut Universitaire de France, Ministère chargé de l'enseignement supérieur, Paris, France. ²⁶Department of Medical Genetics, CHU, Nice, France. ²⁷These authors contributed equally: Alice Rouan, Melanie Pousse, Nadir Djerbi. ✉e-mail: Alice.Rouan@outlook.fr; Eric.Gilson@univ-cote-dazur.fr

