## [Peer Review File · Nature Communications]

Telomere DNA length regulation is influenced by seasonal temperatures differences in short-lived but not in long-lived reef-building coralsREVIEWER COMMENTS

Reviewer #1 (Remarks to the Author):

This paper is impressive and expansive in its scope and integration of transcriptomic data and TRF estimated telomere length of species (both host and symbiont) within two genera across their range of the Pacific basin. While correlational in nature these data provide a powerful test of potential geographic, genera-specific environmental effects on telomere length and gene expression plausibly associated with telomere dynamics (i.e., telomerase, shelterin protein and cell cycle gene expression). It seems evolutionary history (i.e., the evolution of particular life history strategies) has a strong association of telomere length in response to past environmental conditions. The paper is well written and is surprisingly manageable to read given the breadth and complexity of the dataset and analyses performed. It was difficult to find fault in either the molecular or statistical analyses.

I have only minor quibbles. It was mildly annoying that several references by many of the same authors were pre-print and thus not yet peer reviewed. Occasionally, when discussing transcriptomic data, the authors refer to the protein itself being up or down (e.g., LL 436), when only *gene expression* has been measured—an easy fix and easy to slip into that language. More explanation of the Gene Ontology work should be given (exactly how were Go Terms searched and filtered after initial analysis (paper/supplement was sparse on this point)? Why was an alpha 0.05 chosen and not corrected for multiple comparisons—typically there are thousands of differentially upregulated genes, if ever there was a case for correction, it is transcriptomic data).

I found it interesting that cold waves (as apposed to heat waves) had the stronger effect. Why? Given the focus on thermal effects, there are several (~8-12) papers that have investigated thermal effects on telomeres reviewed in "Of telomeres and temperature: measuring thermal effects on telomeres in ectothermic animals" 2021 Mol Ecol* that could be referred to and offer explanations for thermal effects (albeit in vertebrates, but given the focus on humans in the intro, it seems relevant) and consequences of ectothermy and telomere biology more broadly in particular is there anything that other sessile organisms might have in common with corals (e.g., plants? See box in Olsson et al's 2018 paper in Phil Trans*). Does coloniality have anything to do with the lack of correlation with colony size that TL change (or lack of change?). These ideas merit consideration. * I am an author on these two papers, but I am not asking that they be cited themselves, but the work reviewed may be useful fonder for thought. Very cool paper. CRF.

Reviewer #2 (Remarks to the Author):

This study aims to understand the mechanisms (historical and contextual environmental conditions, gene expression) driving divergence in telomere length across populations of two coral genera with contrasting life strategies. For that end, authors measured telomere length using the gold standard assay (TRF) in > 1000 samples of coral colonies distributed across a broad spatial range. This study is highly relevant regarding the need of understanding how telomere lengths are distributed across natural populations coping with contrasting (and potentially risky) environmental conditions. However, although the work is clearly notable, there are some important constrains that limit the conclusions derived from the obtained results. 1) Authors link telomere length with population or individual performance and ageing –this is still under debate and mostly linked to telomere shortening rather than telomere length (and particularly in ectothermic animals). Further information is needed for the studied genera/spp. For example, it is stated that "this finding has implications for the mechanisms regulating stress response and longevity as well as the health and aging patterns of animal and human populations (...)", however, it is unknown the role of coral telomere length in those processes, and authors do not discuss the potential adaptive role of having shorter telomeres under certain scenarios (either linked to the cost of maintaining longer telomeres, or to the adaptive value of having shorter telomeres under certain scenarios; see Casagrande and Hau 2019 in Proceedings of the Royal Society B, and e.g., McLennan et al., 2017 in Functional Ecology in the Atlantic Salmon). Please see also below my comment on the possible use of SMDs which may help in this aspect. 2) Some essential information of colonies is missing.

This includes age: coral colony size is used as a proxy of age, however a likely effect of temperature on growth rate may include confounding effects between growth rate, age, and telomere length. This is also relevant for gene expression as the timing of gene expression and current telomere length may be uncoupled. Also, it is unclear whether colonies were identified at the species level: this is highly relevant to disentangle the effect of environmental conditions and genetic divergence (species across the spatial and environmental gradient?) 3) Authors measured gene expression and found interesting associations between telomere length and some regulatory mechanisms (e.g., the catalytic subunit of the enzyme telomerase, TERT), which add relevant information to the telomere literature. Authors state that "mechanisms regulating the telomere-environment connections have co-evolved with specific life-history traits" but, to the best of my understanding, they did not explore whether the expression of the differentially expressed genes linked to telomere regulation is linked to environmental conditions. 4) I suggest authors to revisit several parts of the manuscript in order to increase its readability which can definitively help to understand better some of their analyses/ideas.

Some other specific comments can be found below:

The role of telomeres in regulating health and ageing is still under debate so please specifically mention it as telomere length does not correlate with lifespan, survival, or fitness in many taxa. Indeed, it is telomere shortening –rather than telomere length- the variable sometimes considered as an indicator of individual ageing and performance.

Line 96: "telomere degradation"?

Line 98: it is also under debate the lack of expression of telomerase after the embryonic stage. For example, it is well known that telomerase is active in somatic tissues of the frog *Xenopus laevis* (e.g., Bousman et al. 2003 in *Journal of Experimental Biology*) or in the bird *Larus michahellis* (Noguera and Velando, 2021 in *Journal of Experimental Biology*). The role of telomerase after birth may be particularly relevant in ectotherms as most of them show indeterminate growth.

104: The redox status is also regulated by multiple genes so telomere dynamics can be indirectly regulated by selection acting on mechanism with the potentiality to increase and reduce the rate of telomere shortening.

116: which findings?

170-171: this is interesting but based on a very few number of samples. From which part of the colony did you collect the samples for telomere analyses?

The introduction does not fully reflect the state-of-the-art of telomere literature. Whereas most of the studies have focused on telomere dynamics within species and/or individual, very little is still known about how telomeres have evolved across populations of a given species, and across taxa. Burraco et al. 2022 in *Ecography* reviewed the available information on telomere dynamics in a spatial context and, to the best of my knowledge, is the first time telomere information has been used for conducting a Species Distribution Model (SDM; see Figure 2 in that paper), which may be considered here to infer a link between telomere length and colony resilience (if data are available).

187-214: I wonder if you know the actual age for some of the studied colonies. Since environmental conditions can lead to differences in growth among populations, and age is assumed to be explained by colony size: how can you disentangle a possible effect of age structure among colonies from environmental effects? And the possible effect of genetic structure/spp? Do you have information about population dynamics within each region?

Please provide the sample size used for testing the link between telomere length and gene expression.

Please provide marginal and conditional R-squared values.

Environmental information used for sPLS analyses is not provided (I think), please provide a full dataset with all the information required to reproduce the analyses.

556-557: A more detailed explanation on how selection model was conducted is needed.

622: The Code availability statement only includes information on the R packages used but does not say anything about the actual R codes used to conduct the analyses.

Do historical/contextual environmental conditions explain variance in genes differentially expressed? (and particularly in those involved in telomere length regulation)

382-384: this sentence does not match with said before in the same paragraph (and see comments above about the role of regulatory mechanisms across the spatial scale in your study system).

389: Please provide more accurate references on the putative role of global change driving telomere dynamics in ectotherms through changes in growth rates

397: "negative correlation" instead of anti-correlation.

404: what do you mean by "telomeres are known to impact stress-response"?

455: What is, in your opinion, the link between your results and climate change? May you generate a variable comparing historical and contextual information in order to see how the change in some environmental variables correlate with telomere length?

I would avoid the use of abbreviation such as T2L or T3L which reduces the readability of the manuscript –and also to reach a broader reader.

Reviewer #3 (Remarks to the Author):

In the presented manuscript, the authors analyzed the length of telomeres, an important marker of genome integrity, accumulated environmental stress, and age/longevity, in two species of stony corals across the Pacific Ocean. This study is remarkable not only for the number of samples collected on a vast geographical area but also by being the first comprehensive study on the biology of telomeres in corals. Corals seem to be great candidates for the study of telomere maintenance as they must have developed strong mechanisms to cope with environmental stress during their exceptionally long lifespan, probably including ways to protect their telomeres from degradation and shortening. Current coral molecular research is strongly focused on understanding the mechanisms that make corals more resilient in the face of the changing climate, and on finding biomarkers of such resilience. The length of telomeres is an obvious choice to study, and I am happy to see it presented here. Moreover, I believe the study of telomere maintenance in corals is of interest to a broad community of scientists as corrupted telomere maintenance is one of the hallmarks of cancer development, premature aging, and senescence. Thus, learning lessons from long-living animals that have clearly figured out how to protect their telomeres over time can be invaluable for human medical field as well.

In my opinion, this is well-done research that examines the role of telomeres and their maintenance in a long-lived resilient *Porites* spp. and rather short-lived more susceptible *Pocillopora* spp. corals from various perspectives and I can't wait to see it published. Anyway, I have the following comments and questions for the team of the authors, in the order as they appear in the text.

Eva Majerova,
Coral Resilience Lab,
Hawai'i Institute of Marine Biology,
UH Mānoa, HI, USA

1. L165-165 I understand why the authors decided to use T2L and T3L abbreviations to stand for “telomere length in the host” and “telomere length in the symbiont” but I am afraid it can be very confusing and hard to understand for the general coral biology audience that is not familiar with the difference in the number of T’s in the telomere repetitions between animals and plants as it is quite a specific knowledge. I would suggest the authors to consider using more clear symbols such as TLh and TLs or anything similar to distinguish between the two.
2. L189-191 “...may be used as a proxy for cumulative growth on colony age.” – please specify if this is based on previous research (provide reference) or if it is your hypothesis.
3. L201-203 I believe that to facilitate reading and understanding the results here, the sentence would benefit if the number of additional samples was mentioned for each site and coral.
4. L210 “{Barkley, 2015 #5126}” looks like untransformed reference.
5. L236 – L238. “For Pocillopora spp. T3Ls, the selected model included island and Symbiodiniaceae communities. In these models, the islands exhibit a unique contribution of 48.1% and 47.3% of the total variance of T2Ls and T3Ls, respectively.” I am not sure I understand this correctly. Why is the first sentence mentioning T3Ls using “model” in singular, while the second one mentions both T2Ls and T3Ls using “these models” in plural?
6. L308-313 I was not able to find appropriate part in the methodology section that would explain the terms “high mean seasonal SST”, “high SST seasonality”, “thermal stress anomaly (TSA) parameters”, “degree heating weeks”, or “degree cooling weeks”. How were they calculated? What data were used to calculate them? It is particularly important to disclose the methods of calculation for DCW and DHW as there were statistically important results observed for these two factors.
7. L313-315 “DHW is a common measure for coral bleaching susceptibility...” please provide reference.
8. L326-327 As much as I understand that a giant project such as the Tara expedition yields in many publications and it is impossible to coordinate their publication time, which results in submitting manuscript with “in preparation, in review, or accepted” references, I think that the result parts referring to these manuscripts need at least some explanation of what was done. It is impossible to review a results part without knowing how the data were acquired. How many samples did the authors analyzed? What method was used for sequencing? What thresholds were set to filter the relevant sequencing data, etc.?
9. L333-336. I had to read this sentence 10 times to finally understand what the authors want to say. Could they try to rewrite it?
10. L411 – L423. This is a very interesting discussion of the presented data in light of other very recent discoveries, and I do not oppose the presented hypothesis. However, unfortunately, it uses only two references to back it up and both of them are currently unpublished manuscripts in the review process. Even though they are available on a preprint server, they are not in my field of expertise, and I don’t feel I can adequately evaluate them.
11. L437-438 “Such a combination of short telomeres with high levels of TERT and TERF gene expression is encountered in some human cancer cells”. While this is true, I am not sure why the authors are mentioning it here. To my knowledge, there is quite firmly established explanation of why cancer cells have short telomeres although the telomere-elongation mechanism is activated. The paper the authors cite here explains: “In the well-established tumorigenesis model, telomeres in human somatic cells gradually become shortened with each cell division. After 50 to 60 cell cycles, cells with shortened telomeres provoke replicative senescence by chromosomal instability and p53 activation, which is induced by the DNA damage response according to telomere shortening [76,77,78]. However, some cells that can overcome senescence by the acquisition of genetic mutations in p53 or other checkpoint proteins continue to proliferate; thus, telomeres become critically short, and apoptosis is subsequently induced (crisis) [79,80]. At this point, a minor population of the cells that activate telomerase (or ALT pathway) acquires immortality and proceeds to carcinogenesis [79].”

Could the authors explain how they see a similarity between coral telomere maintenance and the carcinogenesis-derived maintenance of telomeres in cancer cells?

12. L443-447 “Therefore, even though the TLs of adult coral colonies and human individuals differ in their dependence on chronological age, they oscillate between similar limits, suggesting that, as

in humans, variations in coral TL may have biological significance regarding health, resilience, and longevity". Could the authors explain how they mean this? Pocillopora spp., Porites spp., and humans show the length of telomeres varying in the same range of lengths, but all three have very different longevity range (differing by hundreds of years), different level of resilience (at least against thermal tolerance) and I don't know where we could conclude anything about health assessment in the two coral species.

13. L463-464 "...at least three sampling sites". Is the "sites" the same as "...from 99 reefs..." in the line 174? Or what is the difference between site and reef here?

14. L474 (Hume et al, in preparation) – is this the same reference as in line 415 or is it a different manuscript?

15. L476 – When reading about stripping the membrane and re-hybridizing it with the algal probe, I immediately thought of cross-hybridization issues between the two probes and between the two hybridizations. From my understanding, the two probes do not cross-hybridize as the TL's between the host and the symbiont show very different profiles, is that the author's view as well? And I found the information about stripping the membrane and verifying its efficiency in the reference paper, but I really believe it's worth mentioning somewhere in this manuscript as well to avoid any doubts of the reader.

16. L500-503 "...were calculated for high molecular weight (48.5-15kb) and low molecular weight (10-1 kb)... ..Coefficients were used to transform... using the high molecular weight coefficient for the upper part and the low molecular coefficient for the lower one". I don't think I properly understand this methodology. How did you calculate the coefficients? How did you transform the data accordingly? Also, what happens to the gel zone between 10 – 15 kb?

17. L503-505 "...to be scaled to the closest ladder". Can you provide more information about where the ladders were positioned and how many were there? This is missing from the methodology part and this sentence thus feels incomplete

18. L508 "Intensity was normalized by the size..." How do you normalize by the size? Can you provide the formula?

19. L509 "...intensity above 2 kb were discarded..." Do you mean below 2 kb? Like the lowest part of the gel?

20. L537 How did you incorporate the symbiont composition in the analysis? Did you go for the prevalent type of symbiont? Or the percentage of clades? What were the levels of the factor?

21. L543-545 Why did you use bacterial community composition as an effect here but not in the analysis on the L537?

22. L549-550 "Tara Pacific ITS2 Symbiodiniaceae data release (Version 1) [Data set]. Zenodo. <https://doi.org/10.5281/zenodo.4061796>) Is this an unformatted reference? Or?

23. L573 "Human homologs were identified based on protein sequences. A BLASTp" How did you get coral protein sequences? Why not use Blastx?

We are extremely grateful for the meaningful comments and suggestions of the three reviewers who have enabled us to considerably improve the analysis, interpretation and readability of our manuscript.

Reviewer #1 (Remarks to the Author):

This paper is impressive and expansive in its scope and integration of transcriptomic data and TRF estimated telomere length of species (both host and symbiont) within two genera across their range of the Pacific basin. While correlational in nature these data provide a powerful test of potential geographic, genera-specific environmental effects on telomere length and gene expression plausibly associated with telomere dynamics (i.e., telomerase, shelterin protein and cell cycle gene expression. It seems evolutionary history (i.e., the evolution of particular life history strategies) has a strong association of telomere length in response to past environmental conditions. The paper is well written and is surprisingly manageable to read given the breadth and complexity of the dataset and analyses performed. It was difficult to find fault in either the molecular or statistical analyses. We thank the reviewer for this positive appraisal of our work.

I have only minor quibbles. It was mildly annoying that several references by many of the same authors were pre-print and thus not yet peer reviewed.

We understand and we are sorry for this difficulty. This work is part of a coordinated submission and some of the referred publications are now in revision or already accepted. In fact, we are targeting a common and synchronous publication and to facilitate the work of reviewers, we decided also to provide all submitted/in review/accepted manuscripts in preprint repository to ease the reviewer action. An update of the status of these publications is given in the revised version.

Occasionally, when discussing transcriptomic data, the authors refer to the protein itself being up or down (e.g., LL 436), when only *gene expression* has been measured—an easy fix and easy to slip into that language.

We agree with the reviewer. Thus, we modified the incriminated sentence of the discussion as follows: *“The odd association in Pocillopora spp. between short TLs and increased expression of the telomerase genes could be explained by the concomitant high level of the gene expressing the telomere protein TRF known to behave as a negative regulator of telomere extension by telomerase.”*

More explanation of the Gene Ontology work should be given (exactly how were Go Terms searched and filtered after initial analysis (paper/supplement was sparse on this point)?

The corresponding Methods section has been detailed as follows *“For the human analysis, all GO terms, and their associated genes were gathered from the Ensembl BioMart tool ¹. For the cnidarian analysis, GO terms were predicted from the protein sequences using the InterPro database ² for Pocillopora and Porites separately. GO terms enrichment analysis was performed using the runTest function from the “topGO” R package ³ with a Fisher’s exact test (“fisher” option). The threshold for the p-value was set at 0.05 and no correction for multiple testing was carried out as recommended by the authors.”.*

Why was an alpha 0.05 chosen and not corrected for multiple comparisons—typically there are thousands of differentially upregulated genes, if ever there was a case for correction, it is transcriptomic data).

According to the “topGO” package documentation, multiple correction should not be applied since the method computes the p-value of a GO term conditioned on the neighboring terms. The tests are therefore not independent, and the multiple testing theory cannot apply. Otherwise, 0.05 is a common p-value threshold.

Regarding “differentially upregulated genes”, we would like to draw the attention of the reviewer that we are not comparing the gene expression between two situations. Instead, we used normalized read counts (TPM) for analyzing the links between gene expression and various types of variables, as explained in the Methods section.

I found it interesting that cold waves (as apposed to heat waves) had the stronger effect. Why?

In fact, the negative correlation between TL and cold wave effect is more pronounced for the sTL of the symbionts, both in *Porites* and *Pocillopora*, while the heat wave has an opposite effect on sTL in *Porites* (we adopted the proposition of Reviewers #2/3 to rename, for sake of clarity, T2L by hTL and T3L by sTL). The effect of cold events on coral physiology is a still poorly explored question. A slight contribution of cold wave (much lower than heat waves) to bleaching was reported ⁴, suggesting that cold wave could stress the symbiont. However, the fact that heat wave has an opposite effect on sTL in *Porites*, suggests that the symbiont responds differently to cold and heat waves between *Pocillopora* and *Porites* ⁵. This interesting point is discussed in the revised version as follows: “*It should be noted that Porites spp. sTLs were negatively correlated with cold wave variables including DCW, while heat waves had an opposite effect. A slight cold wave contribution to bleaching has been reported ⁴, suggesting that cold waves may stress the symbiont. However, the fact that the heat waves had an opposite effect on sTL in Porites spp. suggests that the symbiont reacts differently between cold and heat waves and/or when hosted in Pocillopora spp. or Porites spp. ⁵.*”

Given the focus on thermal effects, there are several (~8-12) papers that have investigated thermal effects on telomeres reviewed in “Of telomeres and temperature: measuring thermal effects on telomeres in ectothermic animals” 2021 Mol Ecol* that could be referred to and offer explanations for thermal effects (albeit in vertebrates, but given the focus on humans in the intro, it seems relevant)

We thank the reviewer very much for this suggestion. We believe that the reference is more suitable in the discussion at two locations: “*Moreover, this represents the first field work on the thermal effects on TL variations in aquatic invertebrates ⁶.*” and “*To explain the long TLs in warm islands, one can speculate that these corals live close to their optimum temperature, lowering stress-induced oxidative damage. Analyzing the co-variation of TL and temperature performances in corals might be a useful approach to explore this hypothesis ^{6,7}.*”

and consequences of ectothermy and telomere biology more broadly in particular is there anything that other sessile organisms might have in common with corals (e.g., plants? See box in Olsson et al’s 2018 paper in Phil Trans*).

The parallel, due to their sessility, in term of telomere response to environmental change is interesting and indeed merits to be evoked in the discussion as follows “*Finally, appropriate mechanisms linking telomere response to the environment could be particularly important for sessile organisms, like plants, to counteract the deleterious effects of seasonal variations ^{6,7}.*”

Does coloniality have anything to do with the lack of correlation with colony size that TL change (or lack of change?).

It is difficult to clearly answer this question. Our knowledge on telomere dynamics in colonial organisms remains limited, however there is one report on a colonial ascidian showing both lower telomerase activity and shorter telomeres compared to the offspring ⁸ suggesting a decreased telomere length with age. On the contrary, it was reported that non-colonial asexually reproducing organisms, such as planaria, can maintain or even increase the size of their telomeres during their growth or during regeneration ^{9, 10}. Therefore, we do not believe that maintenance of telomeres in adults is inherent to coloniality. Nevertheless, it is possible that telomere shortening in coral occurred during the early development of *Porites* and *Pocillopora* sp as reported for the coral *Acropora digitifera* ¹¹. For these reasons, we prefer not to discuss specifically a possible role of coloniality in the telomeric dynamics we observed.

These ideas merit consideration.

* I am an author on these two papers, but I am not asking that they be cited themselves, but the work reviewed may be useful fonder for thought.

Very cool paper. CRF.

Reviewer #2 (Remarks to the Author):

This study aims to understand the mechanisms (historical and contextual environmental conditions, gene expression) driving divergence in telomere length across populations of two coral genera with contrasting life strategies. For that end, authors measured telomere length using the gold standard assay (TRF) in > 1000 samples of coral colonies distributed across a broad spatial range. This study is highly relevant regarding the need of understanding how telomere lengths are distributed across natural populations coping with contrasting (and potentially risky) environmental conditions.

We thank the reviewer for his positive appraisal of the interest of our work

However, although the work is clearly notable, there are some important constrains that limit the conclusions derived from the obtained results. 1) Authors link telomere length with population or individual performance and ageing –this is still under debate and mostly linked to telomere shortening rather than telomere length (and particularly in ectothermic animals). Further information is needed for the studied genera/spp. For example, it is stated that “this finding has implications for the mechanisms regulating stress response and longevity as well as the health and aging patterns of animal and human populations (...)”, however, it is unknown the role of coral telomere length in those processes, and authors do not discuss the potential adaptive role of having shorter telomeres under certain scenarios (either linked to the cost of maintaining longer telomeres, or to the adaptive value of having shorter telomeres under certain scenarios; see Casagrande and Hau 2019 in Proceedings of the Royal Society B, and e.g., McLennan et al., 2017 in Functional Ecology in the Atlantic Salmon). Please see also below my comment on the possible use of SMDs which may help in this aspect.

We agree with the reviewer, we need to know the impact of long or short telomeres on coral physiology to fully interpret the consequences of the different TL variation-environment described here. However, due to known functions of telomeres in a wide range of organisms, including ectothermic

animals, it is reasonable to assume that telomere variation has an impact on coral physiology, whether good or bad. To clarify our text regarding this point in the revised version, we : i) add this in the discussion to explicit the limitation of our study : *“If the coral telomeres, like in a wide range of organism, including ectotherms, play a role in stress-response, health, and aging, one can hypothesize that the telomere-environment relationships of Pocillopora spp. and Porites spp. contribute to their differences in stress-resistance and longevity properties...”* : *“”*; ii) add this § in the discussion section : *“It might also be that under protective and unchallenged conditions, the cost of maintaining long telomeres is better supported. If this long TL phenotype might have no other beneficial effect than ensuring chromosome end stability, the short telomeres encountered under stressful conditions may have specific adaptive values, for instance by being more prone to signal survival pathways and cope with energetic trade-offs as was previously proposed ¹²⁻¹⁵. To address the question of the adaptive value of the coral TL variation as a function of environment and island of origin, future studies using population telomere data to predict coral species habitat suitability through species distribution models (SDMs) may be helpful ¹⁶.”*; iii) modify the abstract sentence cited by the reviewer as follows: *« We propose that environmentally regulated mechanisms of telomere maintenance are linked to organismal performances, a matter of paramount importance considering the effects of climate change on health.”*

2) Some essential information of colonies is missing. This includes age: coral colony size is used as a proxy of age, however a likely effect of temperature on growth rate may include confounding effects between growth rate, age, and telomere length.

We agree with the reviewer that the colony size is not a perfect estimate of the age of the colony and that temperature, impacting growth rate, can impact colony size. However, it remains that in natural conditions, by sampling wild untagged colonies, assessing the age of a colony is nearly impossible. For instance, drilling the Porites colonies would have been an environmental disaster. Therefore, we used the colony size as a proxy of the colony age, this is clearly explained and discussed in the text : *“Although coral colony size is considered a poor predictor of genetic age ¹⁷, especially for species prone to frequent breakage and regrowth ¹⁸, it may be used as a proxy for cumulative growth or colony age ¹⁹. Thus, if the TL of the coral colony shortened with cumulative growth or age as in several non-colonial metazoan organisms ^{20, 21}, one would expect a relationship between TL and the size of the colony.”* In addition, please, see below the detailed answer for the lines 187-214.

This is also relevant for gene expression as the timing of gene expression and current telomere length may be uncoupled.

In the revised version, we added a sPLS analysis of gene expression versus environmental variables (new Figure 5): both the temperature at the time of sampling (contextual measures : Sea Surface Temperature or SST, airTemp) and the historical SST regimes (SST-mean, SST, min, SST-max, seasonal-mean, seasonal-min and seasonal-max) were negatively correlated with the expression of telomere genes, in agreement with the sPLS of figure 3 showing a positive-correlation between contextual and historical SST and hTL and Figure 4 showing an negative-correlation between hTL and telomere gene expression. Therefore, the expression of telomere genes varies in the same direction whether it is based on historical SST, contextual SST or TL data, suggesting that the expression of these genes at the time of sampling is coupled to TL variations. This is presented in the Result section of the revised version as *“Both the temperatures at the time of sampling (contextual indicators: SST, airTemp) and the thermal regimes (historical indicators: SST-mean, SST, min, SST-max, seasonal-mean, seasonal-min and seasonal-max) were negatively correlated with the expression of telomere genes (Fig. 5), in*

agreement with the sPLS heatmap of **Fig. 3** showing a positive correlation between contextual and historical thermal indicators and hTL and **Fig. 4** showing a negative correlation between hTL and telomere gene expression. Therefore, the expression of telomere genes varies in the same direction whether it is based on historical or contextual thermal indicators, suggesting that the expression of telomere genes at the time of sampling is coupled to the current TL measures and reflects telomere gene expression pattern over time .”.

Also, it is unclear whether colonies were identified at the species level: this is highly relevant to disentangle the effect of environmental conditions and genetic divergence (species across the spatial and environmental gradient?)

The colonies were collected according to their resemblance to *Porites lobata* or *Pocillopora meandrina*, leading to a collection of closely related species for both genera. The species delimitation (based on genome-wide SNP analysis and diagnostic fragment sequencing) was determined for 11 out of the 32 islands: 3 species for *Porites* sp. (K1-3) and 5 species for *Pocillopora* sp. (SVD1-5) were identified. Thus, the here-found differences hold true beyond the species level and are applicable to the genera *Porites* and *Pocillopora* at large. This is discussed in the revised version as “*These pattern differences (Pocillopora spp. vs. Porites spp.) hold true beyond the species level and are applicable to the genera Porites and Pocillopora at large (see Supplementary Fig. 6b).*”. This is to say that the serendipitous species-level sampling (in hindsight) provided us with independent data to ascertain the observed differences across species boundaries. We therefore would argue that disparate population dynamics across regions would not affect the observed differences. The reviewer is right of course that it would be intriguing to assess whether location-specific telomere differences may align with population demographic rates, which we may explore in related follow-up work. (same answer as given below to a similar comment).

3) Authors measured gene expression and found interesting associations between telomere length and some regulatory mechanisms (e.g., the catalytic subunit of the enzyme telomerase, TERT), which add relevant information to the telomere literature. Authors state that “mechanisms regulating the telomere-environment connections have co-evolved with specific life-history traits” but, to the best of my understanding, they did not explore whether the expression of the differentially expressed genes linked to telomere regulation is linked to environmental conditions.

We thank the reviewer for this interesting suggestion. Therefore, we explored this connection by performing another sPLS analysis. The results are shown in the new Figure 5 of the revised version. The levels of telomere gene expression as a function of environment inversely mirrors the TL variations as a function of environment: high expression in case of high seasonal variability and low expression in hot environment. and heat waves. This is in full agreement with the anti-correlation we observed between telomere gene expression and TL (Figure 4). This is now added to the revised version as follows: “*These results led us to explore whether the expression of the telomere genes is linked to environmental conditions. We explored this connection by performing a sPLS regression. The heatmap based on correlation coefficients obtained via a two-component sPLS model revealed a high expression linked to indicators of high SST seasonal variability and low expression in a warm environment (Fig. 5). This analysis also revealed lower correlation patterns: a positive one with recovery from heat/cold wave indicators and a negative one with heat waves indicators. Pairwise correlations confirmed these results (Supplementary Fig. 12). The plots of the sPLS model revealed a grouping of individual colonies by their island of origin (Supplementary Fig. 13), suggesting that the telomere gene expression pat-*

tern is connected, at least in part, to island effects. This is reminiscent of the TL variation determined by the island of origin (Fig. 2, Supplementary Fig. 7). These results agree with the negative correlation between telomere gene expression and TL (Fig. 4). ”

Regarding “differentially expressed genes”, we would like to draw the attention of the reviewer that we are not using DEG but normalized read counts (TPM) for exploring the links between gene expression and various types of variables, as explained in the Methods section.

4) I suggest authors to revisit several parts of the manuscript in order to increase its readability which can definitively help to understand better some of their analyses/ideas.

Some other specific comments can be found below:

The role of telomeres in regulating health and ageing is still under debate so please specifically mention it as telomere length does not correlate with lifespan, survival, or fitness in many taxa. Indeed, it is telomere shortening –rather than telomere length- the variable sometimes considered as an indicator of individual ageing and performance.

We agree with the referee that the manuscript would benefit from introducing in a more balanced way the wide range of data, sometimes seemingly contradictory, existing between TL and organismal performance. We also agree to stress more about the role of shortening rate. This is now reformulated in the introduction as follows: “A causal relationship between short TL, health and longevity is shown in genetic models of plant, yeast, nematode, killifish, zebrafish, and mouse ²²⁻²⁷. Despite all these findings supporting the view that short TL has deleterious effects, the interspecific variations of lifespan cannot be simply explained by differences in TL ²⁸. Even the concept of the deleterious effects of short TL must be revisited since there are examples of higher survival correlating with short telomeres ^{29, 30}, suggesting the possibility of advantageous effects of short TLs, such as preventing cancer formation or being prone to activate metabolic and survival pathways ^{13, 15, 31}.”

The somewhat disparate range of findings reported above make it necessary to investigate beyond simple TL to understand the mechanisms linking telomeres and organismal performance. This is how it was found that it is rather the rate of TL shortening than absolute mean TL that correlates with lifespan in different organisms ^{20, 32, 33}. Interestingly, in both yeast and mice, there are indications that higher or aberrant rates of telomere erosion increase the percentage of extremely short telomeres ^{34, 35}, which in turn can trigger cellular senescence even when they are in limited numbers ^{36, 37}. Telomeric structures other than TL are certainly also to be taken into consideration, like the presence of a gain-of-function variant of the shelterin TERC1 gene in the long-lived naked mole rat ³⁸. Overall, the impact of telomere regulation on physiology and life-history traits remains a very open question.”

Line 96: “telomere degradation”?

Degradation here refers to the telomere erosion, reflecting end-replication problems occurring at each round of replication as well as nuclease action and recombination. For sake of clarity, we changed it for “erosion” in the revised version.

Line 98: it is also under debate the lack of expression of telomerase after the embryonic stage. For example, it is well known that telomerase is active in somatic tissues of the frog *Xenopus laevis* (e.g., Bousman et al. 2003 in *Journal of Experimental Biology*) or in the bird *Larus michahellis* (Noguera and

Velando, 2021 in Journal of Experimental Biology). The role of telomerase after birth may be particularly relevant in ectotherms as most of them show indeterminate growth.

We thank the reviewer for giving us these references. We agree that we must be clearer on this point. Even in human, telomerase does not completely disappear in somatic cells, particularly in adult stem/progenitor cells, which usually contain a detectable telomerase activity but insufficiently to fully replenish telomere DNA length upon replication. Therefore, in the revised version we change the word “repressed” by “down-regulated” and add more references of organisms where telomerase is maintained or not in adult somatic cells.

104: The redox status is also regulated by multiple genes so telomere dynamics can be indirectly regulated by selection acting on mechanism with the potentiality to increase and reduce the rate of telomere shortening.

Yes, we agree that the role of the large spectrum of genes involved in TL regulation can be direct or indirect, crossing numerous processes. This large number of TL regulating genes offers an almost infinite combination of fine-tuning paths for TL regulation, whether selected or being the corollary of other constraints. Thus, we add ‘directly or indirectly’ in the revised version when referring to the 400 budding yeast genes acting on TL regulation.

116: which findings?

We thank the reviewer for raising this point, which indeed can be confusing. The term ‘findings’ refers collectively to the citations listed in the preceding paragraph illustrating numerous human and non-human studies showing crosstalk between TL, stress, environment, and health. To precise this point and to enlarge the reasoning to the various genetic models of telomere dysfunction, and also to take into account the above comment, we reformulated this part of the introduction (see above).

170-171: this is interesting but based on a very few number of samples. From which part of the colony did you collect the samples for telomere analyses?

The *Pocillopora* branches were chosen randomly in each of the six colonies analyzed, shown in the picture below:

We agree that the number of samples is low. However, we also have unpublished data (to be published elsewhere) showing the same result in another branched coral (*Stylophora pistillata*) cultured in aquaria at the Centre Scientifique de Monaco. Here we took four branches from one colony and cut them into 5 sections of (~2 cm). The corresponding TLs do not show significant difference as can be seen below (see below, for the reviewer only).

In *S. pistillata*, no different TL regarding the sample position on a colony branch (Wilcoxon test, $p > 0.05$) was found.

The introduction does not fully reflect the state-of-the-art of telomere literature. Whereas most of the studies have focused on telomere dynamics within species and/or individual, very little is still known about how telomeres have evolved across populations of a given species, and across taxa. Burraco et al. 2022 in *Ecography* reviewed the available information on telomere dynamics in a spatial context and, to the best of my knowledge, is the first time telomere information has been used for conducting a Species Distribution Model (SDM; see Figure 2 in that paper), which may be considered here to infer a link between telomere length and colony resilience (if data are available).

We thank the reviewer for this advice regarding the existing literature. We modified the introduction accordingly and added other papers that have addressed it since Burraco's synthesis (Fohringer et al. 2022 *BMC Evol. Ecol.* in moose - Karkkainen et al. 2022 *Mol. Ecol.* in flycatchers - Zamora-Camacho et al. 2022 *Science of the Total Environment* in a frog species): "TL differs greatly between taxa, individuals, and ecoregions" ^{16, 31, 39-46} .

Regarding SDM, we agree that it would be very interesting to use our data to compare the telomere-environment connections with the habitability of the studied species. Unfortunately, to the best of our knowledge, data of the presence/absence/density of the targeted genera across the Pacific are

not available. Generating them is possible but will require a very important work of analysis of existing documentation and additional field campaigns that are beyond the scope of this study. The SDM analysis is certainly a very interesting project to be developed, a point discussed as follows “*To address the question of the adaptive value of the coral TL variation as a function of environment and island of origin, future studies using population telomere data to predict coral species habitat suitability through species distribution models (SDMs) may be helpful* ¹⁶.”.

187-214: I wonder if you know the actual age for some of the studied colonies. Since environmental conditions can lead to differences in growth among populations, and age is assumed to be explained by colony size: how can you disentangle a possible effect of age structure among colonies from environmental effects?

There is no simple method to determine the age of the colony in this type of large-scale field study on coral. Therefore, we used colony size as a proxy of cumulative growth and/or age, as explicated in the text. In addition, to avoid any confounding effect of local environment, we made an additional sampling in the same sites of two islands where colonies were collected over a wide range of size, once again no significant relationship was found between colony size and TL, see: “*In order to rule out any confounding effect of the site or island of origin, we sampled additional colonies at the same reef sites according to four diameter classes: one site at Clipperton Island for Pocillopora spp. and Porites spp., and one site for Porites spp. in Palau. No difference in mean T2L among the different classes of colony size was detected (Supplementary Fig. 5c-f).*” Finally, we discuss this point in the manuscript as follows: “*The robustness of this conclusion stems from the large geographical distance and size classes sampled. For instance, the age range of sampled Porites colonies can be estimated between 20 and 600 years assuming an average growth rate of 0.9 cm/year* ⁴⁷. *Moreover, since the Porites spp. colonies are not subjected to frequent fragmentations* ¹⁸, *as confirmed by an absence of genetic clones among a subset of sampled Porites spp. colonies* ⁴⁸, *one expects a good relationship between their size and genetic age.*”

For reviewers only (these data will be published elsewhere): we analyzed TL of another branched coral (*S. Pistillata*) cultured in aquaria and of known (genetic) age, confirming the absence of TL shortening with age. Specifically, we sampled 3 branches, or nubbins (n), of 3 colonies of approximately 1-year-old (A1), 2 colonies of approximately 2 years old (A2), 1 colony of 4 years old (A4) and 2 different mother colonies that were introduced in the Centre Scientifique de Monaco aquarium 30 years old ago (S1 and S2). We considered the group of 1yo, 2yo and 4yo as the young group compared to the old group of colonies from a different genetic background (S1 and S2). See the figure below:

Figure 17: *Stylophora pistillata* of different age. Colonies (C1, C2, C3) of 1-year-old (A1) are displayed on top with orange borders, 2 years old (A2) are shown together with green borders, the only colony of 4 years old (A4) is displayed with purple border and the adult one (S1) with a pink border on the right.

Telomere length was measured twice using TRF assay. We found that the mean host TL doesn't decrease in *S. pistillata* with age. On the opposite, the mean TL of the older colonies (S1 and S2) was significantly higher than the younger ones with a positive slope of ($y=4.5713334 \times T2L+0.0009114$). See Figure below:

Boxplot of Telomere length with age in coral *Stylophora pistillata*. Shapes are representing genet of colonies, colonies are differentiated with colors, each point represents a branch of a colony. Pairwise Wilcoxon test results are displayed (*' $p < 0,05$, '**' $p < 0,01$, '***' $p < 0,001$)

And the possible effect of genetic structure/spp? Do you have information about population dynamics within each region?

The colonies were collected according to their resemblance to *Porites lobata* or *Pocillopora meandrina*, leading to a collection of closely related species for both genera. The species delimitation (based on genome-wide SNP analysis and diagnostic fragment sequencing) was determined for 11 out of the 32 islands: 3 species for *Porites* sp. (K1-3) and 5 species for *Pocillopora* sp. (SVD1-5) were identified. Thus, the here-found differences hold true beyond the species level and are applicable to the genera *Porites* and *Pocillopora* at large. This is discussed in the revised version as “These pattern differences (*Pocillopora* spp. vs *Porites* spp.) hold true beyond the species level and are applicable to the genera *Porites* and *Pocillopora* at large (see **Extended Data Figure 6b**)”. This is to say that the serendipitous species-level sampling (in hindsight) provided us with independent data to ascertain the observed differences across species boundaries. We therefore would argue that disparate population dynamics across regions would not affect the observed differences. The reviewer is right of course that it would be intriguing to assess whether location-specific telomere differences may align with population demographic rates, which we may explore in related follow-up work.

Please provide the sample size used for testing the link between telomere length and gene expression. We include this information in the revised version : “To investigate whether hTL variation is associated with particular patterns of host gene expression, we analysed transcriptomic (RNA-sequencing) data from 55 *Pocillopora* spp. colonies of the subset of 11 islands used for host species assignment ⁴⁹.”

Please provide marginal and conditional R-squared values.

The marginal and conditional R-squared values for the variance partition analyses (linear mixed models) were added as sup tables: Sup Table 2 for Figure 2, Sup Table 4 for Extended Data Figure 6b and Sup Table 8 for Figure 4a.

Environmental information used for sPLS analyses is not provided (I think), please provide a full dataset with all the information required to reproduce the analyses.

It is true that many of the databases used in this manuscript come from other co-submitted articles. Thus, to help the reading of our manuscript, in the revised version, we have grouped for each studied colony, mean hTL, mean sTL and the environmental data used in a single xls file: Sup. Table 7.

555-557: A more detailed explanation on how selection model was conducted is needed.

This has been completed in the revised version as follows: *“For each species and TL, all the possible models were generated, and the best model was selected according to its lowest AIC value”*.

622: The Code availability statement only includes information on the R packages used but does not say anything about the actual R codes used to conduct the analyses.

We added in the revised version that *“the R codes are available upon request.”*

Do historical/contextual environmental conditions explain variance in genes differentially expressed? (and particularly in those involved in telomere length regulation)

An answer comes from the new sPLS we did in response to one of the above comments of the reviewer (new Figure 5) showing that both the historical SST regimens (marked in grey) and the contextual SST measured at the time of sampling (marked in blue) were both negatively correlated with the expression of telomere genes. Therefore, the variation in telomere gene expression as a function of temperature appears similar for historical and contextual data, suggesting that the relative expression of telomere genes at the time of sampling reflects a mechanism explaining TL dynamic over time. This is presented in the revised version as follows : *“Both the temperatures at the time of sampling (contextual indicators: SST, airTemp) and the thermal regimes (historical indicators: SST-mean, SST, min, SST-max, seasonal-mean, seasonal-min and seasonal-max) were negatively correlated with the expression of telomere genes (Fig. 5), in agreement with the sPLS heatmap of Fig. 3 showing a positive correlation between contextual and historical thermal indicators and hTL and Fig. 4 showing a negative correlation between hTL and telomere gene expression. Therefore, the expression of telomere genes varies in the same direction whether it is based on historical or contextual thermal indicators, suggesting that the expression of telomere genes at the time of sampling is coupled to the current TL measures and reflects telomere gene expression pattern over time.”*.

382-384: this sentence does not match with said before in the same paragraph (and see comments above about the role of regulatory mechanisms across the spatial scale in your study system). Within each of the two genera, we were unable to detect a species-specific effect of TL (see Extended Figure 6b), in agreement with the fact that the TL variation is a genus- and not a species-specific effect for the targeted corals. We slightly this sentence to precise this point: *“These pattern differences (Pocillopora spp. vs Porites spp.) hold true beyond the species level and are applicable to the genera Porites and Pocillopora at large (see Supplementary Fig. 6b). Overall, these results unveil that the mechanisms regulating TL in response to environmental conditions exhibit differences between coral genera with different life-history traits. “*

389: Please provide more accurate references on the putative role of global change driving telomere dynamics in ectotherms through changes in growth rates

The goal here is to refer to the possibility that a warmer environment, can be seasonal or caused by global changes, can increase the growth rate of coral: Anderson et al showed for three key coral species that higher growth extension is observed for warmer reefs along the Australian GBR. In the revised version we added another reference showing the same trend for another coral species (Lough, J. M. & Barnes, D. J. Environmental controls on growth of the massive coral *Porites*. *J Exp Mar Biol Ecol* 245, 225–243 (2000)).

397: “negative correlation” instead of anti-correlation.

Corrected

404: what do you mean by “telomeres are known to impact stress-response”?

We refer here to the studies showing that short telomeres can synergize with stressors to induce a large spectrum of deleterious processes altering health and accelerating aging (see for instance ^{50, 51}).

455: What is, in your opinion, the link between your results and climate change? May you generate a variable comparing historical and contextual information in order to see how the change in some environmental variables correlate with telomere length?

The information related to climate change that we can extract from the historical data are heat waves (i.e. the thermal stress anomaly (TSA) parameters and the ‘degree heating weeks’ (DHW), an indicator of warm water thermal stress events). We discussed this point in the revised version as follows “*Surprisingly, for *Pocillopora* spp., a coral genus known to be sensitive to climate change, we did not observe a clear relationship of TLs with heat wave parameters but rather with parameters reflecting seasonal thermal regimen. *Porites* spp., a coral genus known to be more resilient facing climate change, there is a slight effect of heat waves parameters. These results could be explained if *Pocillopora* spp. colonies, in contrast to *Porites* spp. colonies, exhibit a high rate of mortality after heat wave, the stress resistant *Porites* spp. conserving some “TL sequels” of the past heat waves.*”. Although this hypothesis remains speculative, it needs to be considered, for instance in future longitudinal studies encompassing a period of heat waves. Therefore, in the discussion, we emphasize the importance of our findings for future studies on the coral telomeres facing climate change: “*Our findings also suggest that climate change may impact telomere homeostasis in a coral genus-specific manner, a determinant of coral health and biodiversity to consider in future reef restoration interventions.*”.

I would avoid the use of abbreviation such as T2L or T3L which reduces the readability of the manuscript –and also to reach a broader reader.

We thank the reviewer for this remark that matches also with one of reviewer #3. So, as suggested by reviewer #3 we replaced T2L by hTL and T3L by sTL.

Reviewer #3 (Remarks to the Author):

In the presented manuscript, the authors analyzed the length of telomeres, an important marker of genome integrity, accumulated environmental stress, and age/longevity, in two species of stony corals across the Pacific Ocean. This study is remarkable not only for the number of samples collected on a vast geographical area but also by being the first comprehensive study on the biology of telomeres

in corals. Corals seem to be great candidates for the study of telomere maintenance as they must have developed strong mechanisms to cope with environmental stress during their exceptionally long lifespan, probably including ways to protect their telomeres from degradation and shortening. Current coral molecular research is strongly focused on understanding the mechanisms that make corals more resilient in the face of the changing climate, and on finding biomarkers of such resilience. The length of telomeres is an obvious choice to study, and I am happy to see it presented here. Moreover, I believe the study of telomere maintenance in corals is of interest to a broad community of scientists as corrupted telomere maintenance is one of the hallmarks of cancer development, premature aging, and senescence. Thus, learning lessons from long-living animals that have clearly figured out how to protect their telomeres over time can be invaluable for human medical field as well.

In my opinion, this is well-done research that examines the role of telomeres and their maintenance in a long-lived resilient *Porites* spp. and rather short-lived more susceptible *Pocillopora* spp. corals from various perspectives and I can't wait to see it published. Anyway, I have the following comments and questions for the team of the authors, in the order as they appear in the text.

Eva
Coral
Hawai'i
UH

Institute
Mānoa,

Resilience
of
Marine
HI,

Majerova,
Lab,
Biology,
USA

We thank the reviewer for the positive appraisal of our work.

1. L165-165 I understand why the authors decided to use T2L and T3L abbreviations to stand for “telomere length in the host” and “telomere length in the symbiont” but I am afraid it can be very confusing and hard to understand for the general coral biology audience that is not familiar with the difference in the number of T's in the telomere repetitions between animals and plants as it is quite a specific knowledge. I would suggest the authors to consider using more clear symbols such as TLh and TLs or anything similar to distinguish between the two.

We thank the reviewer for this suggestion that we adopted in the revised version. Thus, we chose the abbreviation hTL for the host and sTL for the symbiont.

2. L189-191 “...may be used as a proxy for cumulative growth on colony age.” – please specify if this is based on previous research (provide reference) or if it is your hypothesis.

Yes, colony size as a proxy of age was previously used in several publications. In the revised version, we cite one of them: ¹⁹

3. L201-203 I believe that to facilitate reading and understanding the results here, the sentence would benefit if the number of additional samples was mentioned for each site and coral.

We added the number of samples as follows : “we sampled additional colonies at the same reef sites according to four diameter classes: one site at Clipperton Island for 13 *Pocillopora* spp. colonies and 14 *Porites* spp. colonies, and one site for 12 *Porites* spp. colonies in Palau.”.

4. L210 “⁴⁷” looks like untransformed reference.

This is corrected.

5. L236 – L238. “For Pocillopora spp. T3Ls, the selected model included island and Symbiodiniaceae communities. In these models, the islands exhibit a unique contribution of 48.1% and 47.3% of the total variance of T2Ls and T3Ls, respectively.” I am not sure I understand this correctly. Why is the first sentence mentioning T3Ls using “model” in singular, while the second one mentions both T2Ls and T3Ls using “these models” in plural?

That was a mistake. We changed the text as follows: “For Pocillopora spp. hTLs, the selected model, based on the lowest Akaike information criterion (AIC), included island and Symbiodiniaceae and bacterial community composition in accordance with the variance partition combining a stronger effect of island, with a unique contribution of 48.1% of the total variance of hTL, with a smaller effect of both Symbiodiniaceae and bacterial community composition (Fig. 2).”

6. L308-313 I was not able to find appropriate part in the methodology section that would explain the terms “high mean seasonal SST”, “high SST seasonality”, “thermal stress anomaly (TSA) parameters”, “degree heating weeks”, or “degree cooling weeks”. How were they calculated? What data were used to calculate them? It is particularly important to disclose the methods of calculation for DCW and DHW as there were statistically important results observed for these two factors.

We are sorry for this inconvenience. A complete explanation of these variables is given in the accepted Lombard et al Sci Data publication. Nevertheless, to help the reader we added the following § in the Result section of the revised version :“ To interpret the effect of local seasonal temperature fluctuations and past climatic events, we extracted high resolution (temporal and spatial) satellite sea surface temperature (SST) from the past 14-18 years (see complete methodology in ⁵²). From those, we extracted the mean seasonal variations and calculated the different heatwaves (HW) and cold waves (CW) indices such as the degree heating/cooling weeks (DHW/DCW) and the Thermal Stress Anomaly (TSA) following the Coral Reef Temperature Anomaly Database (CoRTAD) methodology ⁵³ and extracted their frequency, variability and re-recovery from last events. ”

7. L313-315 “DHW is a common measure for coral bleaching susceptibility...” please provide reference. Degree heating week is initially originating from works from ⁵⁴ showing that temperatures exceeding 1 °C above the highest summertime mean temperature are sufficient to cause heat stress that can lead to coral bleaching. Since then, a lot of work have been conducted showing that this stress accumulates through time leading to the calculation of the Degree Heating Week index, commonly used to estimate the cumulated stress experienced by corals. This calculation is commonly used to monitor stress from satellites from various organizations for management purposes (e.g. <https://coralreefwatch.noaa.gov/>, https://ereefs.aims.gov.au/ereefs-aims/gbr1/dhw_heatstress) and is regularly assessed and validated (e.g. <https://link.springer.com/article/10.1007/s00338-009-0502-z>, <https://link.springer.com/article/10.1007/s00338-016-1524-y>) .

We added the three following references to DHW in the revised version: ⁵⁴⁻⁵⁶

8. L326-327 As much as I understand that a giant project such as the Tara expedition yields in many publications and it is impossible to coordinate their publication time, which results in submitting manuscript with “in preparation, in review, or accepted” references, I think that the result parts referring to these manuscripts need at least some explanation of what was done. It is impossible to review a results part without knowing how the data were acquired. How many samples did the

authors analyzed? What method was used for sequencing? What thresholds were set to filter the relevant sequencing data, etc.?

In fact, we are targeting a common and synchronous publication and to facilitate the work of reviewers, we decided also to provide all submitted/in review/accepted manuscripts in preprint repository to ease the reviewer action. This is notably the case for: Lombard et al (*Sci Data* accepted, <https://doi.org/10.1101/2022.05.25.493210>); Belser et al (*Sci Data*, accepted, <https://arxiv.org/abs/2207.02475>); Noel et al <https://doi.org/10.1101/2022.05.17.492263> accepted at *Genome Biology*, Hume et al <https://doi.org/10.1101/2022.10.13.512013> has been reviewed by external reviewers who did not oppose to the general findings

9. L333-336. I had to read this sentence 10 times to finally understand what the authors want to say. Could they try to rewrite it?

We hope that this rephrasing will be clearer: *“All the genes whose expression variation is attributed hTL (except for OSBPL8) do not exhibit an expression variance attributable to the other variables (Supplementary Fig. 8a). Therefore, they can be considered as associated to hTL independently to island, Symbiodiniaceae composition and host species effects.”*

10. L411 – L423. This is a very interesting discussion of the presented data in light of other very recent discoveries, and I do not oppose the presented hypothesis. However, unfortunately, it uses only two references to back it up and both of them are currently unpublished manuscripts in the review process. Even though they are available on a preprint server, they are not in my field of expertise, and I don't feel I can adequately evaluate them.

We are aware of this difficulty for the reviewer, inherent to our strategy of co-publication for the first major results of Tara-Pacific, and we are sorry for it. Notably, the Noel et al manuscript <https://doi.org/10.1101/2022.05.17.492263> is now accepted at *Genome Biology* and the Hume et al paper <https://doi.org/10.1101/2022.10.13.512013> has been reviewed by external reviewers who did not oppose the general findings. Our anticipation is that these two manuscripts will be co-published with the telomere-focused manuscript.

11. L437-438 “Such a combination of short telomeres with high levels of TERT and TERF gene expression is encountered in some human cancer cells”. While this is true, I am not sure why the authors are mentioning it here. To my knowledge, there is quite firmly established explanation of why cancer cells have short telomeres although the telomere-elongation mechanism is activated. The paper the authors cite here explains: “In the well-established tumorigenesis model, telomeres in human somatic cells gradually become shortened with each cell division. After 50 to 60 cell cycles, cells with shortened telomeres provoke replicative senescence by chromosomal instability and p53 activation, which is induced by the DNA damage response according to telomere shortening [76,77,78]. However, some cells that can overcome senescence by the acquisition of genetic mutations in p53 or other checkpoint proteins continue to proliferate; thus, telomeres become critically short, and apoptosis is subsequently induced (crisis) [79,80]. At this point, a minor population of the cells that activate telomerase (or ALT pathway) acquires immortality and proceeds to carcinogenesis [79].”

Could the authors explain how they see a similarity between coral telomere maintenance and the carcinogenesis-derived maintenance of telomeres in cancer cells?

We thank the reviewer for pointing out this possible source of ambiguity in our discussion of telomere gene expression. Our intention here was to provide an example of previously published situations of short TL with elevated expression of telomerase and shelterin expression. It was not a question of drawing a parallel with cancerous cells. So, we agree that the chosen example can be misleading.

In the preceding sentence, we propose a possible mechanism to explain the apparent paradox of an elevated telomerase expression and short TL by an opposite regulation of telomerase by the shelterin subunit TRF. In the case of oncogenesis, other mechanisms are possibly at play, such as the reactivation of telomerase when the telomeres are already very short, determining a new TL equilibrium maintained during the subsequent replications. This situation seems unlikely to us for coral. We rather favor the hypothesis of a need to reinforce telomere stability due to active growth by acting on both replication (telomerase) and protection (shelterin), resulting in a rather short equilibrium TL size. We do not believe it is within the scope of this publication to discuss these hypotheses in detail. Also, to remove any ambiguity on the fact that we are not drawing a parallel with oncogenesis, we have diversified the examples in the revised version by replacing “*Such a combination of short telomeres with high levels of TERT and TERF gene expression is encountered in some human cancer cells* ⁵⁷.” by “*Such a combination of short telomeres with high levels of TERT and/or TERF gene expression is not unprecedented since shorter TLs with high telomerase activity are associated in humans with high allostatic load* ⁵⁸ *while some human cancer cells combine shorter TLs, elevated telomerase activity and high TRF2 expression* ⁵⁷.”.

12. L443-447 “Therefore, even though the TLs of adult coral colonies and human individuals differ in their dependence on chronological age, they oscillate between similar limits, suggesting that, as in humans, variations in coral TL may have biological significance regarding health, resilience, and longevity”. Could the authors explain how they mean this? *Pocillopora* spp., *Porites* spp., and humans show the length of telomeres varying in the same range of lengths, but all three have very different longevity range (differing by hundreds of years), different level of resilience (at least against thermal tolerance) and I don’t know where we could conclude anything about health assessment in the two coral species.

The aim here is to raise the fact that the TL variations observed during aging in humans (but also in zebrafish) and known to have an impact on health and aging are within the same range as the variations observed in corals according to their environment (especially for *Pocillopora*, the TLs of *Porites* varying less). Thus, by analogy with these two vertebrates, one suggests that the size variations observed in coral may have an impact on their physiology. That said, the average length of telomeres in corals is not related to their age, as is encountered in other taxa. One possibility is that it is the shortening rate that is decisive ²⁰, although this hypothesis needs to be substantiated. Nevertheless, we agree with the reviewer that a health assessment of the sampled colonies is needed to test our hypothesis.

To make these points clearer, we slightly changed this § as follows: “*The biological significance of these findings may extend beyond corals. Notably, the hTL differences between colonies (3–10 kb for *Pocillopora* spp. and 3–14 kb for *Porites* spp.) were within the range of TL variation observed in human and zebrafish ageing (4–12 kb) ^{59, 60}. Therefore, even though the TLs of adult coral colonies and human/zebrafish individuals differ in their dependence on chronological age, they oscillate between similar limits, suggesting that, as in these two vertebrates, variations in coral TL may have biological significance regarding health, resilience, and longevity.*”.

13. L463-464 “...at least three sampling sites”. Is the “sites” the same as “...from 99 reefs...” in the line 174? Or what is the difference between site and reef here?

Yes, these sampling sites correspond to the 99 reef sites. This is homogenized in the revised version.

14. L474 (Hume et al, in preparation) – is this the same reference as in line 415 or is it a different manuscript?

Yes, the colony size data are primarily described in Hume et al, which also reports the species delimitation (see <https://doi.org/10.1101/2022.10.13.512013>).

15. L476 – When reading about stripping the membrane and re-hybridizing it with the algal probe, I immediately thought of cross-hybridization issues between the two probes and between the two hybridizations. From my understanding, the two probes do not cross-hybridize as the TL's between the host and the symbiont show very different profiles, is that the author's view as well?

Yes, it's exactly that. This is described in the reference paper (Rouan et al , Mol. Ecol. 2021, see in particular Figure 2).

And I found the information about stripping the membrane and verifying its efficiency in the reference paper, but I really believe it's worth mentioning somewhere in this manuscript as well to avoid any doubts of the reader.

It's true that it is a key point of our methodology. Thus, we have included this sentence in the Results section: *“Efficiency of stripping step between the two probes hybridization was assessed imaging phosphoscreens overnight exposed to stripped membranes⁶¹”*

16. L500-503 “...were calculated for high molecular weight (48.5-15kb) and low molecular weight (10-1 kb)... ..Coefficients were used to transform... using the high molecular weight coefficient for the upper part and the low molecular coefficient for the lower one”. I don't think I properly understand this methodology. How did you calculate the coefficients? How did you transform the data accordingly? Also, what happens to the gel zone between 10 – 15 kb?

To transform the pixel scale into a size scale we used the molecular ladders to build a linear equation between pixel and size to convert the scale. Since higher molecular weights are migrating faster while lower ones are migrating more slowly, we used two different linear equations to extrapolate the size more accurately over the entire gel length. To calculate a ladder linear equation, we reported the pixel position for every band of known size of the ladder (the position of the maximum peak of intensity was chosen for every band). The linear equation from the relation between the pixel and the log(size) was extracted and applied to the entire scale ($\log(\text{size (in bp)}) = a * \text{position (in pixel)} + b$) which leads to the equation ($\text{size} = 2^{(a * \text{position} + b)}$). We used the 3 highest molecular bands (48.5kb, 20kb and 15kb) to extract a linear equation for the high part of the gel and we used the lower bands (10kb, 8kb, 6kb, 5kb, 4kb, 3kb, 2.5kb, 2kb) to extract a linear equation for the lower part. The "high molecular weight" linear equation was applied until 8-10kb. The exact position of the switch between the low and the high molecular weights equation depended on the continuity of the scale and was slightly different between gels but always fell into the (8kb-10kb) range.

In order to clarify this point in the revised version, we added in the revised Method section : *“Since higher molecular weights are migrating faster while lower ones are migrating more slowly, we used two different linear equations to extrapolate the size more accurately over the entire gel length. Fitted linear model coefficients (a,b) of log₂ ladder size (kb) against peak pixel position were calculated with the high molecular weight ladders sizes(48.5kb, 20kb and 15kb) and with the low molecular weight*

ladder sizes (10kb, 8kb, 6kb, 5kb, 4kb, 3kb, 2.5kb, 2kb) using the lm function of “stats” R package. The position of the switch between the low and the high molecular weights equation depended on the continuity of the scale and was slightly different between gels but always fell into an 8kb-10kb range. ”

17. L503-505 “...to be scaled to the closest ladder”. Can you provide more information about where the ladders were positioned and how many were there? This is missing from the methodology part and this sentence thus feels incomplete

We used 2 ladders, the SmartLadder (Eurogenetec) for (10 kb - 0.2 kb) and the QuickLoad 1kb Extend (NEB) for (48.5 kb - 0.5 kb). Both were loaded in following wells at the left end of the gel and the right end of it. In some gels where the number of samples allowed it, the 2 ladders were also put in a middle position of the gel. During the analysis, the lanes were grouped depending on their position in the gel and the closest ladder was used to transform the pixel scale into a kb scale, (left ladders, middle ladders if present, right ladders).

The reference of the two types of ladders was added in the revised method section.

18. L508 “Intensity was normalized by the size...” How do you normalize by the size? Can you provide the formula?

The method was based on ⁶² and the formula is :

$$xnorm_i = \frac{signal\ intensity_i}{size_i}$$

with i being the pixel position.

Which gives the following formula for the calculation of the normalized mean :

$$Mean = \frac{\sum signal\ intensity}{\sum_{i=1}^n \frac{signal\ intensity_i}{size_i}}$$

with n being the total number of pixels.

The reference ⁶² was added in the revised version

19. L509 “...intensity above 2 kb were discarded...” Do you mean below 2 kb? Like the lowest part of the gel?

We apologize for this error in the text and the misunderstanding that arose from it. Yes we meant below 2kb, the lowest part of the gel and was changed accordingly in the text.

20. L537 How did you incorporated the symbiont composition in the analysis? Did you go for the prevalent type of symbiont? Or the percentage of clades? What were the levels of the factor?

In the Method section, it is described : “*Symbiodiniaceae communities were clustered into 47 groups employing partitioning around medoids using Bray-Curtis dissimilarity-generated distances based on square root transformed ITS2 sequence counts (post-MED table ⁶³).*”

21. L543-545 Why did you use bacterial community composition as an effect here but not in the analysis on the L537?

The objective here was to test any species-effect determined for 11 islands. Since the bacterial community composition had null effect at the level of the 32 islands, we did not include it.

22. L549-550 “Tara Pacific ITS2 Symbiodiniaceae data release (Version 1) [Data set]. Zenodo. <https://doi.org/10.5281/zenodo.4061796>) Is this an unformatted reference? Or? Zenodo is referring to a data open repository.

23. L573 “Human homologs were identified based on protein sequences. A BLASTp ...” How did you get coral protein sequences? Why not use Blastx?

We used Blastp instead of Blastx for a matter of time. We had to find homologs for thousands of sequences. In the revised version we now give a detailed description of how these homologs were found : “*Pocillopora* proteins were predicted from the *Pocillopora cf. effusa* genome as described in ⁶⁴. Briefly, CDSs were predicted using the mapping of proteins from 18 Cnidarian species and *Pocillopora cf. effusa* transcripts with Gmove tool ⁶⁵. Based on putative exons and introns from the alignments, Gmove searched for open reading frames (ORFs) consistent with the protein evidence. Finally, putative transposable elements (TEs) were removed as detailed in ⁶⁴. In addition, human homologs of *Pocillopora cf. effusa* proteins were obtained using a BLASTp ⁶⁶ search of the UniProtKB/Swiss-Prot database, restricted to *Homo sapiens*. An e-value threshold of 1e-5 was set to filter the results. For each *Pocillopora cf. effusa* protein, the best hit based on e-value and bit score was selected. ”

1. Kinsella, R.J. *et al.* Ensembl BioMart: a hub for data retrieval across taxonomic space. *Database (Oxford)* **2011**, bar030 (2011).
2. Paysan-Lafosse, T. *et al.* InterPro in 2022. *Nucleic Acids Res* (2022).
3. Alexa, A., Rahnenfuhrer, J. & Lengauer, T. Improved scoring of functional groups from gene expression data by decorrelating GO graph structure. *Bioinformatics* **22**, 1600-1607 (2006).
4. Hoegh-Guldberg, O. & Fine, M. Low temperatures cause coral bleaching. *Coral Reefs* **23**, 444-444 (2004).
5. Pontasch, S. *et al.* Photoacclimatory and photoprotective responses to cold versus heat stress in high latitude reef corals. *J Phycol* **53**, 308-321 (2017).
6. Friesen, C.R., Wapstra, E. & Olsson, M. Of telomeres and temperature: Measuring thermal effects on telomeres in ectothermic animals. *Mol Ecol* (2021).
7. Olsson, M., Wapstra, E. & Friesen, C. Ectothermic telomeres: it's time they came in from the cold. *Philos Trans R Soc Lond B Biol Sci* **373** (2018).
8. Skold, H.N., Asplund, M.E., Wood, C.A. & Bishop, J.D. Telomerase deficiency in a colonial ascidian after prolonged asexual propagation. *J Exp Zool B Mol Dev Evol* **316**, 276-283 (2011).
9. Tan, T.C. *et al.* Telomere maintenance and telomerase activity are differentially regulated in asexual and sexual worms. *Proc Natl Acad Sci U S A* **109**, 4209-4214 (2012).
10. Garcia-Cisneros, A. *et al.* Long telomeres are associated with clonality in wild populations of the fissiparous starfish *Coscinasterias tenuispina*. *Heredity (Edinb)* **115**, 480 (2015).
11. Tsuta, H., Shinzato, C., Satoh, N. & Hidaka, M. Telomere shortening in the colonial coral *Acropora digitifera* during development. *Zoolog Sci* **31**, 129-134 (2014).
12. Young, A.J. The role of telomeres in the mechanisms and evolution of life-history trade-offs and ageing. *Philos Trans R Soc Lond B Biol Sci* **373** (2018).
13. Casagrande, S. & Hau, M. Telomere attrition: metabolic regulation and signalling function? *Biol Lett* **15**, 20180885 (2019).

14. McLennan, D. *et al.* Habitat restoration weakens negative environmental effects on telomere dynamics. *Mol Ecol* (2021).
15. Jacome Burbano, M.S. & Gilson, E. The Power of Stress: The Telo-Hormesis Hypothesis. *Cells* **10** (2021).
16. Burraco P, Lucas P & P, S. Telomeres in a spatial context: a tool for understanding ageing pattern variation in wild populations. *Ecography*, e05565 (2022).
17. Hughes, T.P. & Jackson, J.B. Do corals lie about their age? Some demographic consequences of partial mortality, fission, and fusion. *Science* **209**, 713-715 (1980).
18. Devlin-Durante, M.K., Miller, M.W., Caribbean Acropora Research, G., Precht, W.F. & Baums, I.B. How old are you? Genet age estimates in a clonal animal. *Mol Ecol* **25**, 5628-5646 (2016).
19. Williams, A.D., Brown, B.E., Putchim, L. & Sweet, M.J. Age-Related Shifts in Bacterial Diversity in a Reef Coral. *PLoS One* **10**, e0144902 (2015).
20. Whittemore, K., Vera, E., Martinez-Navado, E., Sanpera, C. & Blasco, M.A. Telomere shortening rate predicts species life span. *Proc Natl Acad Sci U S A* **116**, 15122-15127 (2019).
21. Remot, F. *et al.* Decline in telomere length with increasing age across nonhuman vertebrates: A meta-analysis. *Mol Ecol* (2021).
22. Watson, J.M. & Riha, K. Telomeres, aging, and plants: from weeds to Methuselah - a mini-review. *Gerontology* **57**, 129-136 (2011).
23. Folgueras, A.R., Freitas-Rodriguez, S., Velasco, G. & Lopez-Otin, C. Mouse Models to Disentangle the Hallmarks of Human Aging. *Circ Res* **123**, 905-924 (2018).
24. Carneiro, M.C., de Castro, I.P. & Ferreira, M.G. Telomeres in aging and disease: lessons from zebrafish. *Dis Model Mech* **9**, 737-748 (2016).
25. Harel, I. *et al.* A platform for rapid exploration of aging and diseases in a naturally short-lived vertebrate. *Cell* **160**, 1013-1026 (2015).
26. Kupiec, M. Biology of telomeres: lessons from budding yeast. *FEMS Microbiol Rev* **38**, 144-171 (2014).
27. Ahmed, S. & Hodgkin, J. MRT-2 checkpoint protein is required for germline immortality and telomere replication in *C. elegans*. *Nature* **403**, 159-164 (2000).
28. Monaghan, P. Telomeres and life histories: the long and the short of it. *Ann N Y Acad Sci* **1206**, 130-142 (2010).
29. McLennan, D. *et al.* Shorter juvenile telomere length is associated with higher survival to spawning in migratory Atlantic salmon. *Funct. Ecology* **31**, 2070-2079 (2017).
30. Wood, E.M. & Young, A.J. Telomere attrition predicts reduced survival in a wild social bird, but short telomeres do not. *Mol Ecol* **28**, 3669-3680 (2019).
31. D'Angiolo, M. *et al.* Telomeres are shorter in wild *Saccharomyces cerevisiae* isolates than in domesticated ones. *Genetics* (2022).
32. Bize, P., Criscuolo, F., Metcalfe, N.B., Nasir, L. & Monaghan, P. Telomere dynamics rather than age predict life expectancy in the wild. *Proc Biol Sci* **276**, 1679-1683 (2009).
33. Boonekamp, J.J., Mulder, G.A., Salomons, H.M., Dijkstra, C. & Verhulst, S. Nestling telomere shortening, but not telomere length, reflects developmental stress and predicts survival in wild birds. *Proc Biol Sci* **281**, 20133287 (2014).
34. Chang, M., Arneric, M. & Lingner, J. Telomerase repeat addition processivity is increased at critically short telomeres in a Tel1-dependent manner in *Saccharomyces cerevisiae*. *Genes Dev* **21**, 2485-2494 (2007).
35. Vera, E., Bernardes de Jesus, B., Foronda, M., Flores, J.M. & Blasco, M.A. The rate of increase of short telomeres predicts longevity in mammals. *Cell Rep* **2**, 732-737 (2012).
36. Hemann, M.T., Strong, M.A., Hao, L.Y. & Greider, C.W. The shortest telomere, not average telomere length, is critical for cell viability and chromosome stability. *Cell* **107**, 67-77 (2001).

37. Abdallah, P. *et al.* A two-step model for senescence triggered by a single critically short telomere. *Nat Cell Biol* **11**, 988-993 (2009).
38. Augereau, A. *et al.* Naked mole rat TRF1 safeguards glycolytic capacity and telomere replication under low oxygen. *Sci Adv* **7** (2021).
39. Haussmann, M.F., Vleck, C.M. & Nisbet, I.C. Calibrating the telomere clock in common terns, *Sterna hirundo*. *Exp Gerontol* **38**, 787-789 (2003).
40. Heidinger, B.J. *et al.* Telomere length in early life predicts lifespan. *Proc Natl Acad Sci U S A* **109**, 1743-1748 (2012).
41. Cook, D.E. *et al.* The Genetic Basis of Natural Variation in *Caenorhabditis elegans* Telomere Length. *Genetics* **204**, 371-383 (2016).
42. Hansen, M.E. *et al.* Shorter telomere length in Europeans than in Africans due to polygenetic adaptation. *Hum Mol Genet* **25**, 2324-2330 (2016).
43. Wilbourn, R.V. *et al.* Age-dependent associations between telomere length and environmental conditions in roe deer. *Biol Lett* **13** (2017).
44. Karkkainen, T. *et al.* Population differences in the length and early-life dynamics of telomeres among European pied flycatchers. *Mol Ecol* **31**, 5966-5978 (2021).
45. Fohringer, C. *et al.* Large mammal telomere length variation across ecoregions. *BMC Ecol Evol* **22**, 105 (2022).
46. Zamora-Camacho, F.J., Burraco, P., Zambrano-Fernandez, S. & Aragon, P. Ammonium effects on oxidative stress, telomere length, and locomotion across life stages of an anuran from habitats with contrasting land-use histories. *Sci Total Environ*, 160924 (2022).
47. Barkley, H.C. *et al.* Changes in coral reef communities across a natural gradient in seawater pH. *Sci Adv* **1**, e1500328 (2015).
48. Hume, B.C.C. *et al.* Disparate patterns of genetic divergence in three widespread corals across a pan-Pacific environmental gradient highlights species-specific adaptation trajectories. *BioRxiv* <https://doi.org/10.1101/2022.10.13.512013> (2022).
49. Armstrong, E. *et al.* Transcriptomic plasticity and symbiont shuffling underpin *Pocillopora* acclimatization across heat-stress regimes in the Pacific Ocean. *Nature Communications*, in revision doi.org/10.1101/2021.11.12.468330 (2022).
50. Rudolph, K.L. *et al.* Longevity, stress response, and cancer in aging telomerase-deficient mice. *Cell* **96**, 701-712 (1999).
51. Chakravarti, D., LaBella, K.A. & DePinho, R.A. Telomeres: history, health, and hallmarks of aging. *Cell* **184**, 306-322 (2021).
52. Lombard, F. & al., e. Open science resources from the Tara Pacific expedition across coral reef and surface ocean ecosystems. *Scientific Data*, in press doi:10.1101/2022.05.25.493210 (2022).
53. Saha, K. *et al.* The Coral Reef Temperature Anomaly Database (CoRTAD) Version 6 - Global, 4 km Sea Surface Temperature and Related Thermal Stress Metrics from 1982 to 2021. *NOAA National Centers for Environmental Information* <https://doi.org/10.25921/ffw7-cs39> (2018).
54. Glynn, P. & D'Croz, L. Experimental evidence for high temperature stress as the cause of El Niño coincident coral mortality. *Coral Reefs* **8**, 181-191 (1990).
55. Sully, S., Burkepale, D.E., Donovan, M.K., Hodgson, G. & van Woesik, R. A global analysis of coral bleaching over the past two decades. *Nat Commun* **10**, 1264 (2019).
56. Voolstra, C.R. *et al.* Standardized short-term acute heat stress assays resolve historical differences in coral thermotolerance across microhabitat reef sites. *Glob Chang Biol* **26**, 4328-4343 (2020).
57. Okamoto, K. & Seimiya, H. Revisiting Telomere Shortening in Cancer. *Cells* **8** (2019).
58. Zalli, A. *et al.* Shorter telomeres with high telomerase activity are associated with raised allostatic load and impoverished psychosocial resources. *Proc Natl Acad Sci U S A* **111**, 4519-4524 (2014).

59. Henriques, C.M., Carneiro, M.C., Tenente, I.M., Jacinto, A. & Ferreira, M.G. Telomerase is required for zebrafish lifespan. *PLoS Genet* **9**, e1003214 (2013).
60. Aubert, G., Baerlocher, G.M., Vulto, I., Poon, S.S. & Lansdorp, P.M. Collapse of telomere homeostasis in hematopoietic cells caused by heterozygous mutations in telomerase genes. *PLoS Genet* **8**, e1002696 (2012).
61. Rouan, A. *et al.* Telomere dysfunction is associated with dark-induced bleaching in the reef coral *Stylophora pistillata*. *Mol Ecol* **31**, 6087-6099 (2022).
62. Li, B. & de Lange, T. Rap1 affects the length and heterogeneity of human telomeres. *Mol Biol Cell* **14**, 5060-5068 (2003).
63. Hume, B.C.C. *et al.*, Tara Pacific ITS2 Symbiodiniaceae data release version 1. *Zenodo* doi:10.5281/zenodo.4061797 (2020).
64. Noel, B. *et al.*, Pervasive tandem duplications and convergent evolution shape coral genomes. *Genome Biology, in press* <https://doi.org/10.1101/2022.05.17.492263> (2022).
65. Dubarry, M. *et al.* Gmove a tool for eukaryotic gene predictions using various evidences. *F1000Research* <https://doi.org/10.7490/f1000research.1111735.1> (2016).
66. Altschul, S.F., Gish, W., Miller, W., Myers, E.W. & Lipman, D.J. Basic local alignment search tool. *J Mol Biol* **215**, 403-410 (1990).

REVIEWERS' COMMENTS

Reviewer #1 (Remarks to the Author):

The authors have adequately addressed all of my comments/suggestions. Thank you.

Reviewer #2 (Remarks to the Author):

This is a revised version of the manuscript "Telomere DNA length regulation is influenced by seasonal temperature differences in short-lived but not long-lived reef-building corals". As I said in my previous revision, this is a great contribution to the field and the work conducted is impressive and novel. In this revised version (and after a very well-performed response to reviewers' comments), authors have added new analyses on the effect of environmental conditions on gene expression linked to TL, which I do think has improved the relevance and impact of the study. Also, I consider that, overall, authors have enhanced the readability of the manuscript, now including an easier-to-follow approach to telomere biology, and the relationship between contrasting environmental conditions, life-history traits, and telomere dynamics. At this point, my main concern is still the one expressed in my previous report, linked to the difficulty to disentangle the possible relevance of genetic divergence or speciation on the environmental effect on coral telomere lengths. Just to re-explain this point: since this study includes samples collected along thousands of km across the Pacific ocean, environmental conditions might have selected by different species or genotypes with different telomere lengths and/or different life histories, such as changes in growth rate or age structure. Said this, authors did efforts to distinguish between "island" and "species" effect via the study of the variation in TL across both "factors" in a subset of islands and species (see comment below). It is also a bit unclear the provided mechanistic explanation for the observed patterns: while they point out that colony size, stated as a proxy of age, is not very important for telomere length in the studied genera, they suggest that growth rate may be behind some of the observed patterns. However, growth rate is normally understood as the gain in body mass per amount of time, and age is stated not to affect coral telomere lengths. In my opinion, these gaps of knowledge are normal, but should be clearly indicated throughout the manuscript (e.g., this cross-sectional and field study cannot fully disentangle whether the association between telomere length and environmental conditions is a cost paid by individuals, or a result of the adaptation to different thermal regimes). There are some other relatively minor aspects that I still consider they may include to clarify and support several parts of the study (please, see below):

L81: Note that TTAGGG sequence seems not to be present across all vertebrates (<https://www.biorxiv.org/content/10.1101/2022.03.25.485759v1.full>). Also, it is important to highlight that telomere shortening (and not telomere length) is considered the "hallmark" of ageing.

L82: Telomere functions do not depend on "the number of these repeats". That sentence suggests that the number of repeats determine the function of telomeres, which to the best of my knowledge is not true.

I suggest adjusting the y-axis to the same values for each genera in Fig1a

189: I do not think an explained variation of 9% by colony size can be considered weak in this case

208: do you know how is the correlation between age and growth across a coral lifetime? Is this expected to be different across the two studied genera? Even in ectotherms with indeterminate growth, an asymptote is often found at certain age, although this can be species and context dependent.

232: I wonder why you selected the model with the lowest AIC instead of doing model averaging (e.g., those models with a difference in AIC lower than 2).

The use of TL information from known species and across island is a good effort in order to disentangle the putative effect of genetic or species effect on the obtained results. However, I wonder how analyses cope with the low number of islands available for some species.

How accessible are the raw climate data and methodology used in this study?

424: why does this correlation suggest a coupling between cell division and telomeres? Also: colony size was not indicative of telomere length for *Porites* spp. I suggest checking some papers on temperature, growth rate and telomere shortening in ectotherms that can help in this regard:
<https://onlinelibrary.wiley.com/doi/full/10.1111/mec.15888>
<https://onlinelibrary.wiley.com/doi/abs/10.1111/gcb.15305>
<https://onlinelibrary.wiley.com/doi/full/10.1111/mec.13857>
<https://royalsocietypublishing.org/doi/full/10.1098/rstb.2016.0449>

It would be great to clarify across the manuscript the time period (14-18 years) used to estimate the effect of historical thermal conditions on TL. Since these genera differ in their life-history traits, the use of ~16-year data means that you have information for a very different % of their lifetime. What "seasonal" means should be clarified a bit more.

Do you have information on other water parameters such as pH or turbidity?

448-466: I fully agree that a possible divergent evolution of regulatory mechanisms is worth exploring. However: a causal relationship between lifespan and stress response must be experimentally determined, plus TL does not clearly differ between the two genera.

478-489: did you find any correlation between telomere maintenance genes and others potentially linked to telomere shortening? I wonder if, as you suggest, the overexpression of some of those maintenance mechanisms is linked to an allostatic response due the overexpression of other mechanisms potentially eroding telomeres. Otherwise, this may indicate that we need to explore much further the mechanisms underlying telomere regulation, which is exciting.

573-574: have you further explored the link between environmental conditions/gene expression and Q1 and Q3 telomeres?

Pablo Burraco -note that I do not expect authors to cite any of my publications.

Reviewer #3 (Remarks to the Author):

I would like to thank the authors for their very thorough answers to all my comments and to the comments of other fellow reviewers. I think the manuscript has significantly improved after the initial reviewing round and I now fully support its publication. I'm looking forward to being able to share these results with my colleagues.

Reviewer #1 (Remarks to the Author):

The authors have adequately addressed all of my comments/suggestions. Thank you.

we thank the reviewer for his very useful comments to improve our manuscript

Reviewer #2 (Remarks to the Author):

This is a revised version of the manuscript “Telomere DNA length regulation is influenced by seasonal temperature differences in short-lived but not long-lived reef-building corals”. As I said in my previous revision, this is a great contribution to the field and the work conducted is impressive and novel. In this revised version (and after a very well-performed response to reviewers' comments), authors have added new analyses on the effect of environmental conditions on gene expression linked to TL, which I do think has improved the relevance and impact of the study. Also, I consider that, overall, authors have enhanced the readability of the manuscript, now including an easier-to-follow approach to telomere biology, and the relationship between contrasting environmental conditions, life-history traits, and telomere dynamics. At this point, my main concern is still the one expressed in my previous report, linked to the difficulty to disentangle the possible relevance of genetic divergence or speciation on the environmental effect on coral telomere lengths. Just to re-explain this point: since this study includes samples collected along thousands of km across the Pacific ocean, environmental conditions might have selected by different species or genotypes with different telomere lengths and/or different life histories, such as changes in growth rate or age structure.

We thank the reviewer for his thoroughness and conscientiousness, pointing out some of our current knowledge gaps. We considered these comments in the revised version as it follows.

Said this, authors did efforts to distinguish between “island” and “species” effect via the study of the variation in TL across both “factors” in a subset of islands and species (see comment below).

We agree that we cannot completely separate "species" from "island" effects, larger scale and genetically resolving studies coupled with controlled aquarium studies will be needed to provide a definitive conclusion on this important point. In the Result section of the revised version, we leave this point open with “Nevertheless, some contribution of host species to TL cannot be ruled out, as suggested by the fact that SVD4 had similar hTLs among different islands (Coiba, Las Perlas, Ducie and Gambier Islands).”

It is also a bit unclear the provided mechanistic explanation for the observed patterns: while they point out that colony size, stated as a proxy of age, is not very important for telomere length in the studied genera, they suggest that growth rate may be behind some of the observed patterns. However, growth rate is normally understood as the gain in body mass per amount of time, and age is stated not to affect coral telomere lengths. In my opinion, these gaps of knowledge are normal, but should be clearly indicated throughout the manuscript (e.g., this cross-sectional and field study cannot fully disentangle whether the association between telomere length and environmental conditions is a cost paid by individuals, or a result of the adaptation to different thermal regimes).

We added the sentence suggested by the reviewer in the following context of the Discussion section :

“If this long TL ...tance by being more prone to signal survival pathways and cope with energetic trade-offs as was previously proposed^{26, 47, 48, 89}. Nevertheless, this cross-sectional

and field study cannot fully disentangle whether the association between TL and environmental conditions is a cost paid by individuals, or a result of the adaptation to different thermal regimes. Thus, to address the question of the adaptive...

There are some other relatively minor aspects that I still consider they may include to clarify and support several parts of the study (please, see below):

L81: Note that TTAGGG sequence seems not to be present across all vertebrates (<https://www.biorxiv.org/content/10.1101/2022.03.25.485759v1.full>).

Many thanks for this information. We changed the introduction accordingly by deleting “all”.

Also, it is important to highlight that telomere shortening (and not telomere length) is considered the “hallmark” of ageing.

We make this point clear in the introduction section as it follows: “*it is rather the rate of TL shortening than absolute mean TL that correlates with lifespan in different organisms*”.

L82: Telomere functions do not depend on “the number of these repeats”. That sentence suggests that the number of repeats determine the function of telomeres, which to the best of my knowledge is not true.

The goal of this sentence is to emphasize that telomeric chromatin structure (and thus key telomere functions) depends on the number of repeats, an important functional point for a study exploring the of TL variation. For instance, the formation of t-loop depends on the number of repeats (Griffith et al, Cell 1999) and in budding yeast, a controlled reduction of TL at a single chromosome end can impact telomerase activity (Marcand et al, EMBO, 1999) and senescence (Abdallah et al, NCB 2009). However, we agree that telomere functions do not only depend on TL and its folding into chromatin, but also of the nature and combination of subtelomeric sequences (chromosome-specific effects) as well as of trans-acting factors (e.g. the level of shelterin subunits). So, for sake of clarity, in the revised version we replaced “*telomere function*” by “*telomere-specific chromatin structure*”

I suggest adjusting the y-axis to the same values for each genera in Fig1a
Done

189: I do not think an explained variation of 9% by colony size can be considered weak in this case

We prefer to keep “weak” in the title of this chapter since no correlation was found in a dedicated sampling at Clipperton. Nevertheless, we deleted “weak” line 209 when referring to the 9% variation.

208: do you know how is the correlation between age and growth across a coral lifetime? Is this expected to be different across the two studied genera? Even in ectotherms with indeterminate growth, an asymptote is often found at certain age, although this can be species and context dependent.

According to Buddemeier & Kinzie (1976 Coral growth. Oceanography and Marine Biology Annual Review 14, 183–225), the short-lived *Pocillopora meandrina* shows determinate growth but the long-lived, massive *Porites* shows indeterminate growth with no systematic decrease in skeletal extension over decades.

232: I wonder why you selected the model with the lowest AIC instead of doing model

averaging (e.g., those models with a difference in AIC lower than 2). We decided to use a selection model procedure rather than model averaging since in those analyses we were more interested in knowing which variables have an impact on telomere length rather than estimating them. In addition, due to the very few models with a $\Delta AIC < 2$, performing model averaging would not have been relevant in our case

The use of TL information from known species and across island is a good effort in order to disentangle the putative effect of genetic or species effect on the obtained results. However, I wonder how analyses cope with the low number of islands available for some species.

We agree that this analysis is only based on a subset of 11 islands. This limitation is taken into account in the result section with “*Nevertheless, some contribution of host species to TL cannot be ruled out, as suggested by the fact that SVD4 had similar hTLs among different islands (Coiba, Las Perlas, Ducie and Gambier Islands)*” and in the discussion with the sentence suggested by the reviewer : “*Nevertheless, this cross-sectional and field study cannot fully disentangle whether the association between TL and environmental conditions is a cost paid by individuals, or a result of the adaptation to different thermal regimes*”.

How accessible are the raw climate data and methodology used in this study?

The complete description of environmental data methodology and the access to the corresponding dataset will be published in Lombard, F. *et al*, Open science resources from the Tara Pacific expedition across coral reef and surface ocean ecosystems. *Scientific Data*, in press and already accessible at doi:10.1101/2022.05.25.493210 (2022), see also <https://zenodo.org/record/6299409#.YkRwrTdBzjA>.

In the revised version, the data availability statement is the following : “*The Telomere DNA length (TL) data generated in this study have been deposited in the Zenodo database under accession code doi.org/10.5281/zenodo.3999999ZZ. The other variables used in this work are available as follows: environmental parameters (historical and contextual): <https://www.biorxiv.org/content/10.1101/2022.05.25.493210v1.full.pdf> and <https://zenodo.org/record/6299409#.YkRwrTdBzjA>; RNAseq: doi.org/10.5281/zenodo.7398767; symbiodinaceae (ITS2) community: <https://doi.org/10.5281/zenodo.4061796>; bacterial (16S rRNA) community: <https://doi.org/10.5281/zenodo.4073268>; colony size: doi.org/10.1101/2022.10.13.512013; species delimitation: doi.org/10.1101/2022.10.13.512013 and <https://doi.org/10.1101/2022.10.21.513203>. Source data are provided with this paper.*”

424: why does this correlation suggest a coupling between cell division and telomeres? Also: colony size was not indicative of telomere length for *Porites* spp.

At this stage, no unequivocal interpretation can be drawn. The purpose of this paragraph is to discuss possible explanations for the longer *Pocillopora* TLs found in warmer islands. A first possibility we examined is that growth (known to be temperature-stimulated in corals) increases TL. This hypothesis was based on a previous report showing that TLs were longer in the regenerating arms than in the non-regenerating arms of starfish (Garcia-Cisneros et al, *Heredity*, 2015, 115, 437-443), suggesting that stimulated cell division can lead to TL elongation in this invertebrate. Additionally, there is evidence that certain environmental conditions can lead to longer TLs: alcohol in yeast (Romano et al. *PLoS Genet*, 2013, 9, e100372) and spaceflight in humans (Luxton et al, *Cell Rep.* 2020 Dec 8;33(10):108435). However, the fact that TL is negatively correlated with size in *Pocillopora* contradicts this hypothesis and instead reflects a TL-shortening effect of cell division, an interpretation supported by RNAseq analysis

performed in 11 islands. Thus, we concluded that increased growth cannot simply explain the longer telomeres of *Pocillopora* spp. in the warmer islands.

Thus, in the revised manuscript, we clarify our reasoning and now cite the starfish publication as follows:

*“Since warmer temperature can increase coral growth {Anderson, 2017 #5127;Lough, 2000 #5287} and, in starfish, arm regeneration is associated to longer TL {Garcia-Cisneros, 2015 #5281}, warmer seasonal conditions could lengthen TL due to increased cumulative growth. However, the opposite was observed, i.e. a tendency for hTL shortening in the apex of *Pocillopora* spp. branches where coral tissue expands (**Supplementary Fig. 1**) and in larger *Pocillopora* spp. colonies (**Supplementary Fig. 5a**), indicating that coral growth is associated with TL shortening rather than elongation in *Pocillopora* spp..”*

A I suggest checking some papers on temperature, growth rate and telomere shortening in ectotherms that can help in this regard:

<https://onlinelibrary.wiley.com/doi/full/10.1111/mec.15888>

<https://onlinelibrary.wiley.com/doi/abs/10.1111/gcb.15305>

<https://onlinelibrary.wiley.com/doi/full/10.1111/mec.13857>

<https://royalsocietypublishing.org/doi/full/10.1098/rstb.2016.0449>

We thank the reviewer for this bibliographic advice. We agree that comparing, across a variety of ectothermic species, the relationships between growth rate, cumulative growth and TL is an exciting topic that certainly merits further investigation. We believe that such a meta-analysis is beyond the scope of the discussion of this manuscript.

It would be great to clarify across the manuscript the time period (14-18 years) used to estimate the effect of historical thermal conditions on TL. Since these genera differ in their life-history traits, the use of ~16-year data means that you have information for a very different % of their lifetime.

Regarding the historical data, please, note that the full methodology is present in very a condensed manner in the present manuscript and in a more detailed manner in the accepted manuscript available here (<https://www.biorxiv.org/content/10.1101/2022.05.25.493210v1>, Lombard et al, Scientific Data, in press) and in an even more detailed manner on the source dataset available here (<https://zenodo.org/record/6499374#.ZBQ5nezMKEA>). Data have been recovered from the satellite MODIS Aqua and Terra launches (2002) with addition of VIIRS-SNPP from 2012 to the date of sampling of Tara pacific (between 2016 to 2018)). The correct timing for the raw data sets is 14 to 16 years and is now corrected in the revised manuscript.

If the raw data have different length, for technical reasons, we used indices that are independent from the length of the time series. For this we followed the well-established CORTAD method <https://www.ncei.noaa.gov/access/metadata/landing-page/bin/iso?id=gov.noaa.nodc:NCEI-CoRTADv6>. This covers immediate conditions at the sampling dates ("snapshot" data in the zenodo dataset), seasonal signals extractions such as maximal/minimal seasonal temperatures (and "snapshot" anomalies at the sampling date from this mean seasonal signal), extraction of several "heat waves and cold waves" indicators (intensity and frequency, calculated only on the past 12 weeks for immediate indicators, on the last 52 weeks for frequency indicators). All other indicators are proposed as mean/std/min and max values to indeed be independent of the time period considered (i.e. reflecting the historical intensity, frequency and variability of temperature variations). Additional metrics of the last heating and cooling events as well as the time of recovery since this latter is also provided to represent the state of thermal stress at the day of sampling.

We are therefore confident that all past climatic indicators used here (because calculated to be independent on the time period considered) are only reflecting the average seasonal signals and the variability of deviations from this seasonal signal (either at the date of sampling, on a defined period prior to sampling or as an average variability indices) and are therefore valid in this context. For sake of clarity, we added in this revised version of the manuscript: *“Thus, if the raw data have different length, for technical reasons, we used historical indicators that are independent from the length of the time series. We are therefore confident that they are only reflecting the average seasonal signals and the variability of deviations from this seasonal signal and are therefore valid to compare coral genera with different life-history.”*

What “seasonal” means should be clarified a bit more.

Mean seasonal signal is defined as the mean-smoothed temperature variations over a mean year. In more detail (see <https://www.biorxiv.org/content/10.1101/2022.05.25.493210v1> and datasets): each time series was first averaged over year-days (julian days: one mean over every years for each calendar dates). The average was triplicated and concatenated into 3 consecutive identical year day averages. A digital low pass filter (filter order 3, pass band ripple 0.1; filfilt function in matlab) with 36 days windows was applied to the concatenated time series to remove high frequency noise. The seasonal cycle was then extracted from the concatenated time series as its middle year. For sake of clarity, we added in the revised version *“Of note, mean seasonal signal is defined as the mean-smoothed temperature variations over a mean year.”*

Do you have information on other water parameters such as pH or turbidity? “

We did measured various "immediate" measurements during the sampling (all mentioned in the accepted manuscript: <https://www.biorxiv.org/content/10.1101/2022.05.25.493210v1>), but see also supp Table 6 here listing all "historical" and immediate "Envir contextual" datasets that were extracted for the samples used in this manuscript. However, those latter are only snapshots measurements and does not reflect past stress over coral (only immediate constraints). . Please, note that we actually tested all potential data available and kept only relevant (and significant) ones for the manuscript (thanks to the sPLS analysis).

In the current version only on *Pocillopora* did pH have a very minor effect (we used measured source measurements of total Alkalinity "corrected_AT" and total carbonates "corrected_CT" both corrected at 20°C for inter-comparison; but see here <https://doi.pangaea.de/10.1594/PANGAEA.944420> for the full dataset explanation and availability and <https://www.biorxiv.org/content/10.1101/2022.05.25.493210v1> for a global explanation) (Figure 3).

Turbidity was not measured directly but thanks to separated variables measuring total particulate organic carbon (mergedPOC) which does not appear as important in our analysis and only a very indirectly related variable (fluorescence of colored dissolved organic matter [fCDOM]) did got again only a weak correlation with TL of *Porites* samples.

Giving the high preponderance of temperature variability indices in our analysis we therefore focused mostly on these indicators, even if some additional more "immediate" measurements did get finally selected through the analysis (all the ones without coloured symbols in Figure 3).

448-466: I fully agree that a possible divergent evolution of regulatory mechanisms is worth exploring. However: a causal relationship between lifespan and stress response must be experimentally determined, plus TL does not clearly differ between the two genera.

We agree that the connection between stress resistance and longevity must be experimentally addressed, this is why we write “.....in stress-resistance and longevity properties”.

478-489: did you find any correlation between telomere maintenance genes and others potentially linked to telomere shortening? I wonder if, as you suggest, the overexpression of some of those maintenance mechanisms is linked to an allostatic response due the overexpression of other mechanisms potentially eroding telomeres. Otherwise, this may indicate that we need to explore much further the mechanisms underlying telomere regulation, which is exciting.

We agree that the RNAseq results point out that further efforts should be made to identify the genes involved the coral TL response to environment. In this respect, it is interesting that the expression of 6 genes encoding proteasome subunits (out of 33 : *PSMB 2,5,7* and *PSMC1,2,6*) is positively correlated with TL and therefore anti-correlated with cell division (Supplementary Figure 10), suggesting an ubiquitin-based regulation of TL maintenance in response to environment. Further combined aquarium-field studies should tell us about the biological significance of these correlations.

573-574: have you further explored the link between environmental conditions/gene expression and Q1 and Q3 telomeres?

No, the analyses presented in this publication were performed with the mean TL. We agree that testing whether Q1 or Q3 exhibit a specific pattern would be interesting.

Pablo Burraco -note that I do not expect authors to cite any of my publications.

Reviewer #3 (Remarks to the Author):

I would like to thank the authors for their very thorough answers to all my comments and to the comments of other fellow reviewers. I think the manuscript has significantly improved after the initial reviewing round and I now fully support its publication. I'm looking forward to being able to share these results with my colleagues.

We thank the reviewer for her very useful comments to improve our manuscript